# Estuarine plastisphere as an overlooked source of N$_2$O production

Xiaoxuan Su[1,2,12], Leyang Yang[1,3,12], Kai Yang [1], Yijia Tang [4], Teng Wen[5], Yingmu Wang[6], Matthias C. Rillig[7,8], Lena Rohe[9], Junliang Pan [10], Hu Li[1] & Yong-guan Zhu [1,3,11✉]

"Plastisphere", microbial communities colonizing plastic debris, has sparked global concern for marine ecosystems. Microbiome inhabiting this novel human-made niche has been increasingly characterized; however, whether the plastisphere holds crucial roles in biogeochemical cycling remains largely unknown. Here we evaluate the potential of plastisphere in biotic and abiotic denitrification and nitrous oxide (N$_2$O) production in estuaries. Biofilm formation provides anoxic conditions favoring denitrifiers. Comparing with surrounding bulk water, plastisphere exhibits a higher denitrifying activity and N$_2$O production, suggesting an overlooked N$_2$O source. Regardless of plastisphere and bulk water, bacterial and fungal denitrifications are the main regulators for N$_2$O production instead of chemodenitrification. However, the contributions of bacteria and fungi in the plastisphere are different from those in bulk water, indicating a distinct N$_2$O production pattern in the plastisphere. These findings pinpoint plastisphere as a N$_2$O source, and provide insights into roles of the new biotope in biogeochemical cycling in the Anthropocene.

[1] Key Laboratory of Urban Environment and Health, Institute of Urban Environment, Chinese Academy of Sciences, 1799 Jimei Road, 361021 Xiamen, China. [2] Interdisciplinary Research Center for Agriculture Green Development in Yangtze River Basin, College of Resources and Environment, Southwest University, 400715 Chongqing, China. [3] University of the Chinese Academy of Sciences, 100049 Beijing, China. [4] School of Life and Environmental Sciences, The University of Sydney, Sydney, NSW 2015, Australia. [5] School of Geography, Nanjing Normal University, 210023 Nanjing, China. [6] College of Civil Engineering, Fuzhou University, 350116 Fuzhou, China. [7] Freie Universität Berlin, Institute of Biology, Berlin, Germany. [8] Berlin-Brandenburg Institute of Advanced Biodiversity Research, Berlin, Germany. [9] Thünen Institute of Climate-Smart Agriculture, Bundesallee 65, Berlin, Germany. [10] School of Electrical Engineering, Chongqing University, 400044 Chongqing, China. [11] State Key Laboratory of Urban and Regional Ecology, Research Center for Eco-Environmental Sciences, Chinese Academy of Sciences, 100085 Beijing, China. [12] These authors contributed equally: Xiaoxuan Su, Leyang Yang. ✉email: ygzhu@rcees.ac.cn

Plastic debris is ubiquitous and problematic in global marine ecosystems[1–3]. With the increasing public perception of plastic pollution, increasing research efforts have been devoted to revealing the impacts of plastic debris on fish[4], invertebrates[5], sea birds[6], turtles[7], and microorganisms[8] in oceans. In recent years, these buoyant plastic debris pieces in marine environments have been reported to provide durable substrates for microorganisms, facilitating colonization and biofilm formation[9,10]. Microbial communities on plastic debris have been called "plastisphere" and represent a novel feature in the Anthropocene, even sometimes called the eighth continent, sparking global interest[9–14]. Recent studies have shown that plastisphere exhibited great differences in microbial community and assembly from surrounding bulk water[12,14]. Further, plastisphere microbial communities could contribute to the acquisition and spread of pathogens and antibiotic resistance in aquatic environments[15]. These works suggest that the plastisphere is indeed capable of inducing negative effects at the ecosystem levels. However, the crucial roles of the plastisphere in biogeochemical cycling in marine environments are largely unknown.

Estuaries are the transitional areas between inland waters and remote oceans[16], acting as hot zones for biogeochemical cycling. An estimated 1.2–2.5 million tons of plastic debris enter estuarine and coastal regions from inland rivers and lakes annually[17]. Therefore, any influence of plastics and the associated plastisphere on estuarine ecology and biogeochemical cycling could be particularly important. In addition to plastic contamination, estuaries are also subjected to unprecedented reactive nitrogen loads from anthropogenic activity, which leads to water eutrophication and ecological perturbations in estuaries[18,19]. Under anoxic conditions, microbially mediated denitrification in estuarine waters, stepwise reductions of nitrate or nitrite to potent greenhouse gas $N_2O$ or further to $N_2$[20], is the major process to eliminate the reactive nitrogen[21]. This pathway can alleviate the effects of eutrophication[21,22], but might contribute to global warming and ozone depletion[23]. Therefore, attention on denitrification and associated $N_2O$ productions should be paid in eutrophic estuarine regions. In the presence of plastic debris, microbial biomass in the plastisphere is likely considerable[10], and anoxic microenvironments would occur during biofilm formation[24,25]. Thus, the emerging plastisphere could harbor great but unrecognized potential in nitrogen removal and $N_2O$ production. Elucidating the role of plastisphere in denitrification, especially in eutrophic waters, is of great significance for water quality remediation and climate change evaluation.

Research on denitrification processes in terrestrial and aquatic environments mainly focuses on bacterial denitrification (BD)[22,26,27]. More evidence has shown that fungal denitrification (FD) and chemodenitrification (CD) could also contribute to nitrogen removal and formation of $N_2O$[28–34]. Recent studies suggest that a high abundance of fungi, such as parasitic and saprophytic ones, could be attached to the plastic surfaces in aquatic environment[35]. Moreover, fungal denitrifiers generally lack $N_2O$ reductase, leading to $N_2O$ as the end product[36,37]. Thus, fungal denitrifiers may represent a potent contributor to $N_2O$ production relative to bacterial denitrifiers (reduction of nitrate to $N_2$ as the end product). In addition, in organic matter- and iron-rich environments, iron (II) reacts with the reactive nitrogen to increase $N_2O$ production from abiotic denitrification[38–40], which may also be a great contributor relative to BD. In fact, the importance of FD and CD in nitrogen transformation and $N_2O$ production has been recognized across terrestrial ecosystems[31,38,41]. However, their contributions in marine waters are largely unknown. Comparing to BD, whether FD and CD processes have similar importance for nitrogen removal and $N_2O$ yield in marine plastisphere requires further investigation.

Here, we choose four types of plastics and establish in situ and lab-scale incubations in an estuary of Xiamen, China to investigate BD, FD, and CD potential in the plastisphere and to differentiate the contributions of these processes to $N_2O$ production (experimental design, see Supplementary Fig. 1). We hypothesize that (i) the surface of plastic debris could be a site of plastisphere formation that provides the necessary conditions for denitrification, (ii) plastisphere has higher denitrifying activity than surrounding bulk water, and (iii) FD and CD processes may have a great contribution to $N_2O$ production relative to BD. To test these hypotheses, we first measure messenger bis(3′–5′)-cyclic dimeric guanosine monophosphate) (c-di-GMP), extracellular polymeric substances (EPS), and intracellular lipid/fatty acid levels to explore the plastisphere formation. Then the denitrifying activities in plastisphere and bulk water are detected by $^{15}N$ isotope-pairing technique, and the relative contributions of BD, FD, and CD to total $N_2O$ production are estimated using $N_2O$ isotopocules analysis. Finally, the keystone bacterial and fungal denitrifiers of plastisphere and bulk water are identified. Our results fill a knowledge gap regarding biogeochemical cycling in the plastisphere and indicate that the new plastic niche may be a potential source of $N_2O$ emission under global environmental change.

## Results

**Biofilm structure and activity**. After the 30-d incubation in the estuary, the four types of plastic debris and surrounding bulk water were harvested. Using scanning electron microscope (SEM), we found that microorganisms in water were attached to the plastics and formed dense microbial clusters (Supplementary Fig. 2), suggesting biofilm formation on the surface of plastics.

We further explored anoxic microbial activities in the plastisphere and bulk water (Fig. 1). When cultured with heavy water ($D_2O$), assimilation of D by metabolically active cells into new biomass can be explicitly and quantitatively detected by Raman as a new C–D band (Fig. 1a). Thus the ratio of (C–D)/(C–D + C–H) is a quantitative indicator for microbial phenotypic activity[15,42]. We acquired the Raman spectrograms of 40–50 microbial cells in each plastisphere and bulk water after 30 days, and calculated their (C–D)/(C–D + C–H) ratios. The ratios in each plastisphere (mean values 0.18–0.29) showed no statistical variations (Tukey, $P = 0.203–0.953$, Fig. 1b), but were greater than those in bulk water (mean value 0.09) ($P < 0.001$). This indicated a higher phenotypic activity in the plastisphere than in bulk water.

**Biofilm composition, c-di-GMP, and lipid/fatty acid signals**. A schematic diagram of microbial colonization processes on plastic surfaces in aquatic environments is illustrated in Fig. 2a. We detected the EPS contents (i.e., extracellular polysaccharide (PS), protein (PN), and DNA (eDNA)) in both the plastisphere and bulk water. As expected, the concentrations of PS, PN, and eDNA were all significantly greater than those in bulk water ($P < 0.001$, Fig. 2b–d). Except for PN and eDNA in the PVC plastisphere, no significant difference ($P = 0.125–0.669$) was detected among each plastic debris (Fig. 2b–d).

The intracellular second messenger c-di-GMP is associated with biofilm formation[43]. We observed higher concentrations of c-di-GMP in the plastisphere (0.013–0.019 μg mg⁻¹ protein) than in bulk water (below detection limit) ($P < 0.001$, Supplementary Fig. 3). This showed that high levels of c-di-GMP commonly support microbial growth in a sessile mode, whereas the low level could result in the planktonic mode[25]. There were no obvious differences among the plastisphere formed by different plastic types ($P = 0.135–0.987$, Supplementary Fig. 3), suggesting that plastic types could not influence microbial c-di-GMP levels.

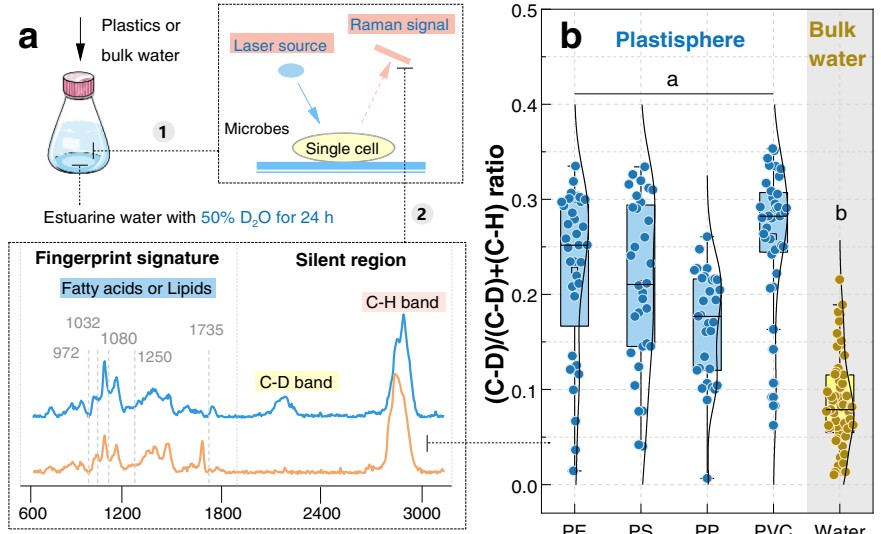

**Fig. 1 D$_2$O-labeled single-cell Raman spectroscopy showing microbial metabolic activities under anoxic conditions in the plastisphere and bulk water.**
**a** Schematic diagram of the D$_2$O-labeled single-cell Raman technique. 1 represents the incubation and preparation of microorganism liquids; 2 represents Raman spectroscopy. Raman characteristic bands assigned to intracellular lipid/fatty acid biomolecules are pinpointed. **b** The ratios of (C–D)/(C–D + C–H) calculated from 40 to 50 single cells in the plastisphere and bulk water. Each point is a single cell. Different letters (a and b) indicate the significant differences ($P < 0.05$) among each plastisphere group and bulk water group.

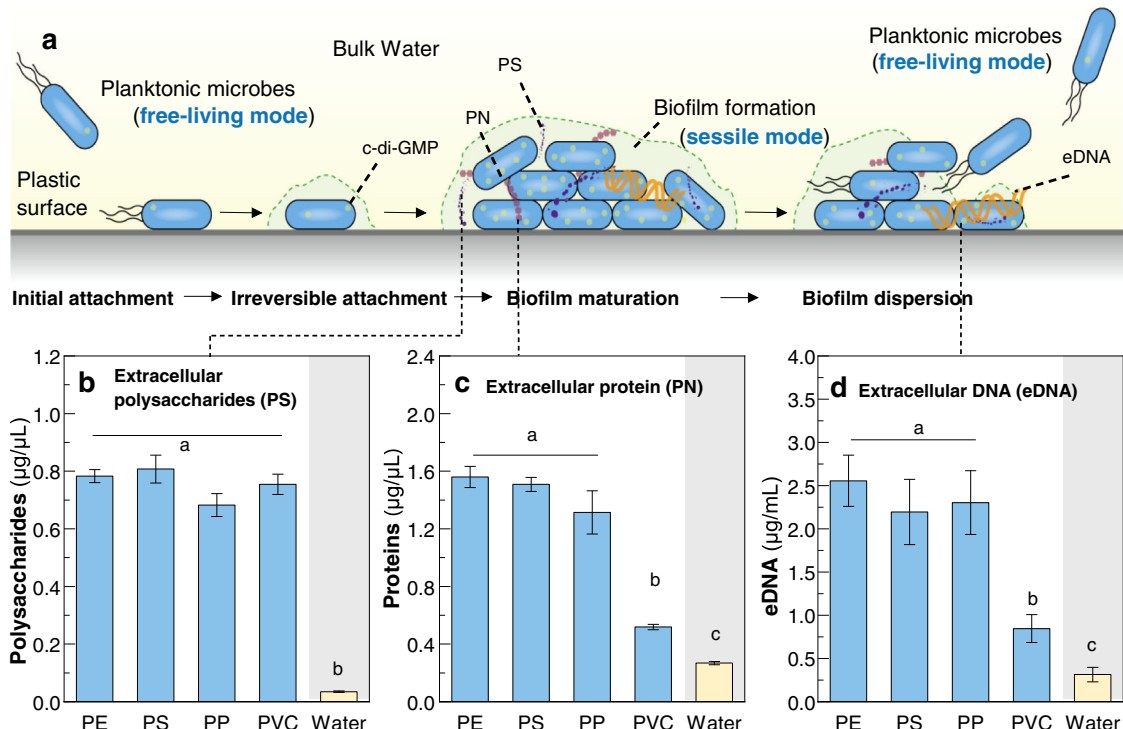

**Fig. 2 Concentrations of extracellular polymeric substances (EPS) in the plastisphere and bulk water. a** Schematic overview of biofilm formation and dispersion on plastic surfaces in water, including cell initial attachment, irreversible attachment, maturity, and dispersion. This diagram is modified from previous studies[25,48]. **b–d** Concentrations of extracellular polysaccharide (PS), extracellular protein (PN), and extracellular DNA (eDNA). Error bars are the standard error. Different letters (a, b, and c) indicate the significant differences ($P < 0.05$) among each plastisphere group and bulk water group.

Microbial lipid and fatty acid levels may also be involved in biofilm formation and dispersion[44]. Using D$_2$O-labeled single-cell Raman spectroscopy, a total of five characteristic bands assigned to lipid or fatty acids (i.e., 972, 1032, 1080, 1250, and 1735 cm$^{-1}$) were identified (Fig. 3). Among the five bands, the plastisphere had no striking difference in peak intensities with bulk water at the bands of 972 and 1735 cm$^{-1}$ ($P = 0.268$–$0.883$, Fig. 3a, e). At 1032, 1080, and 1250 cm$^{-1}$, however, the peak intensities in the plastisphere were lower than those in bulk water ($P < 0.001$, Fig. 3b–d), highlighting the different lipid/fatty acid levels in microorganisms between the plastisphere and bulk water.

**Denitrification and N$_2$O production pathway.** During the in situ incubation, the plastic debris was suspended in the water

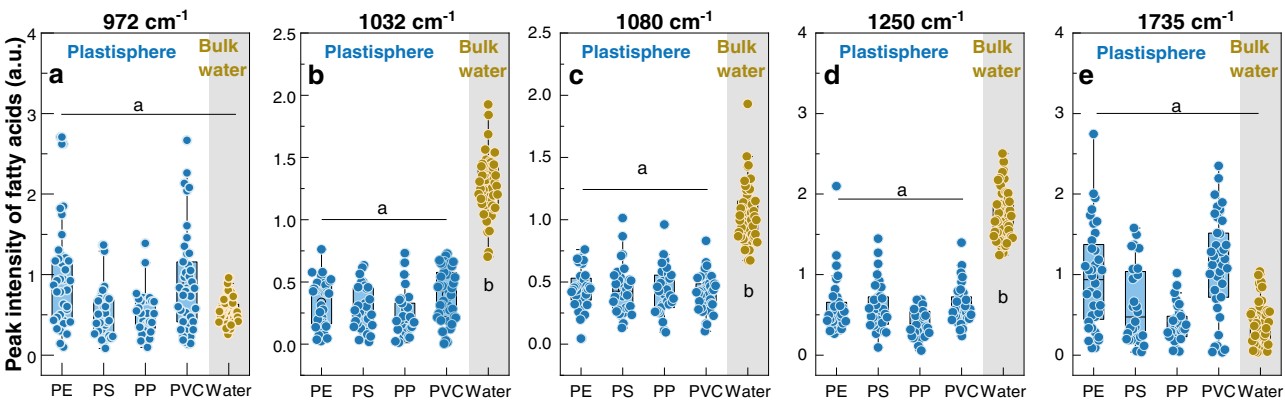

**Fig. 3 Raman profiling of intracellular lipid/fatty acid levels in microorganisms of the plastisphere and bulk water. a** 972 cm$^{-1}$. **b** 1032 cm$^{-1}$. **c** 1080 cm$^{-1}$. **d** 1250 cm$^{-1}$. **e** 1735 cm$^{-1}$. Different letters (a and b) indicate the significant differences ($P < 0.05$) among each plastisphere group and bulk water group.

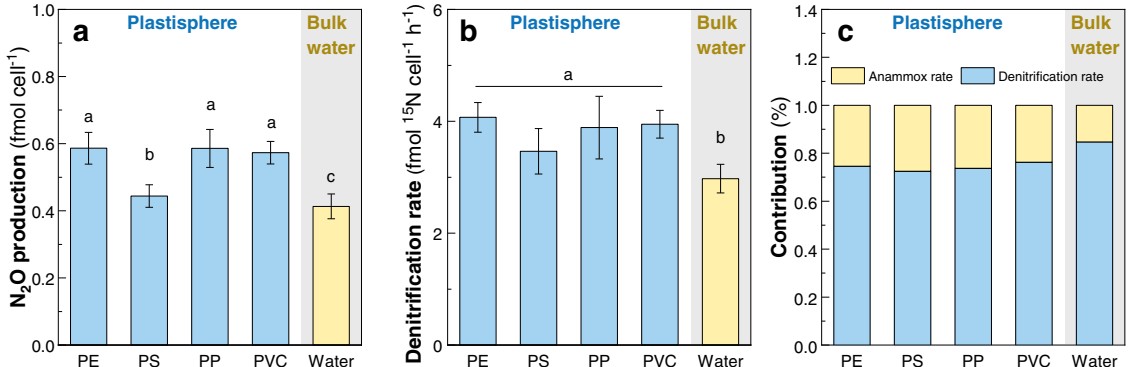

**Fig. 4 Denitrifying activity and N$_2$O production in the plastisphere and bulk water. a** Concentrations of N$_2$O after the 24 h denitrification experiment. **b** Denitrification rates quantified by $^{15}$N isotope-pairing technique. Both N$_2$O production and denitrification rate are normalized with microbial cell numbers by using the absolute abundances of 16S rRNA in the plastisphere and bulk water (see "Methods"). **c** Contributions of denitrification and Anammox processes to N$_2$ production. Error bars are the standard error. Different letters (a, b, and c) indicate the significant differences ($P < 0.05$) among each plastisphere group and bulk water group.

depth of ~4 m, where the dissolved oxygen (DO) concentration was 1.19 mg L$^{-1}$ (<2 mg L$^{-1}$, generally considered as hypoxia[45]). Thus, we focused on the denitrification-derived N$_2$O in the lab-scale experiments of this study. Plastisphere and bulk water showed pronounced differences in denitrification rate and N$_2$O production (Fig. 4). After 24 h, N$_2$O concentrations were 0.44–0.59 fmol cell$^{-1}$ in the plastisphere, greater than 0.41fmol cell$^{-1}$ in bulk water ($P < 0.001$–0.042, Fig. 4a). N$_2$O concentrations in the PS plastisphere were lower than the other types of plastisphere ($P < 0.001$–0.012). For denitrification rate, the plastisphere registered higher rates ranging from 3.36 to 4.07 fmol cell$^{-1}$ h$^{-1}$, while the bulk water had a lower rate (mean value 2.97 fmol cell$^{-1}$ h$^{-1}$) ($P = 0.001$, Fig. 4b). These corroborated that the plastisphere could be a potential source of denitrification and N$_2$O production. Comparing with Anammox process, denitrification contributed 73–84% of total N$_2$ productions (Fig. 4c), meaning the dominant role of denitrification in nitrogen removal in the estuary.

We further measured N$_2$O isotope values in the plastisphere and bulk water (Supplementary Fig. 4). Coupling N$_2$O-δ$^{18}$O with N$_2$O-SP (site preference, see "Methods") values, we could investigate the N$_2$O dynamics during denitrification. In the plastisphere, N$_2$O-δ$^{18}$O were 64.14–77.82‰ and N$_2$O-SP were 15.56–26.21‰, while they were 55.62–59.93‰ and 7.53–8.68‰ in bulk water, respectively ($P < 0.001$, Supplementary Fig. 4), signifying a shift in N$_2$O production pathways. After calibration for oxygen exchange, a multiple-isotope plot (N$_2$O/H$_2$O-δ$^{18}$O vs

N$_2$O-SP) was generated (Fig. 5a). Considering N$_2$O mixing and reduction, we established a N$_2$O isotope-based mass balance model to differentiate the relative fractions of BD, FD, and CD processes to N$_2$O productions (see "Methods"). We found that BD was the major N$_2$O source, contributing 51.08–67.06% and 77.10–79.59% of total N$_2$O production in the plastisphere and bulk water, respectively ($P = 0.008$, Fig. 5b). FD unexpectedly contributed 8.55–46.40% of total N$_2$O production, and the contribution in the plastisphere was higher than that in bulk water ($P < 0.001$, Fig. 5b). These demonstrated that the plastisphere exhibited a different N$_2$O production pattern from bulk water. Comparing with BD and FD, the proportion of the CD process (0–10.28%) can be ignored, for both plastisphere and bulk water.

**Denitrifier communities.** We first quantified the abundances of 16S rRNA and ITS genes in the plastisphere and bulk water (Supplementary Fig. 5), and found that the abundances of bacteria and fungi were higher in the plastisphere than those in bulk water ($P < 0.001$–0.027). Denitrifier communities were then investigated by high-throughput sequencing based on three denitrifying marker genes (i.e., bacterial *nirS* (b*nirS*), bacterial *nirK* (b*nirK*), fungal *nirK* (f*nirK*)). As indicated by α-diversity, the plastisphere had a higher diversity and richness than bulk water ($P < 0.05$, Supplementary Fig. 6). The diversity of b*nirS*-type denitrifiers reached the highest, followed by f*nirK*-type and

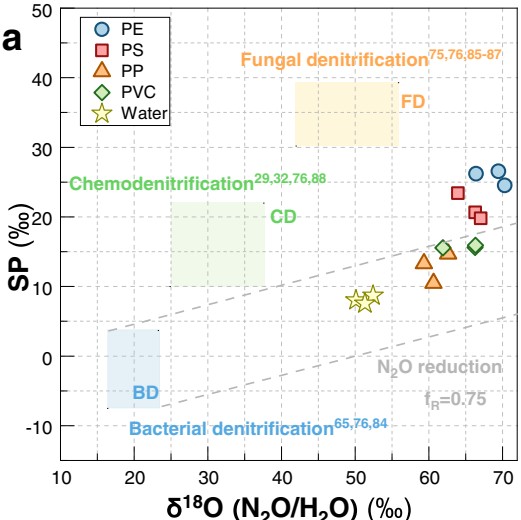
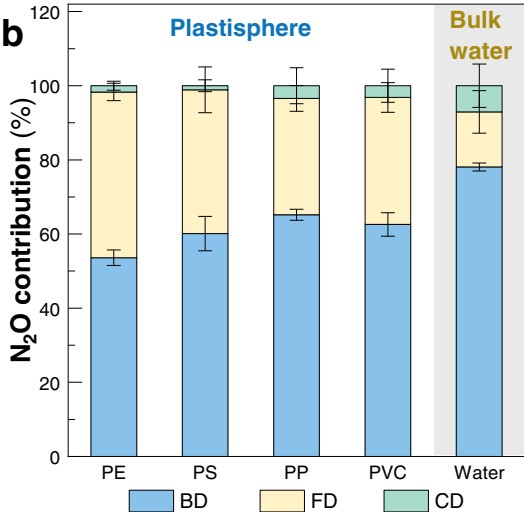

**Fig. 5 Multiple-isotope signatures of N₂O illustrating the relative contributions of bacterial denitrification (BD), fungal denitrification (FD), and chemodenitrification (CD) in the plastisphere and bulk water. a** Relationships between N₂O-SP and N₂O-$\delta^{18}$O (N₂O/H₂O) values in each group. N₂O-$\delta^{18}$O values were corrected for H₂O-$\delta^{18}$O assuming almost complete O exchange. Ranges of putative isotope signatures of N₂O ($\delta^{18}$O (N₂O/H₂O)) from BD, FD, and CD are indicated as the three color boxes. The corresponding references are listed in the figure. The area enveloped by gray dash lines represents the N₂O reduction through BD process based on fractionation effects (slope=SP$_{\varepsilon N2O}$/$\delta^{18}$O$_{\varepsilon N2O}$ = 0.25). This mapping approach is modified from the method in ref. [83]. **b** Relative proportions of N₂O production via BD, FD, and CD in the plastisphere and bulk water. Errors were estimated by Monte Carlo simulation and reflected the accuracy of N₂O isotope mass balance model. Although the contributions of each denitrification process would shift with the prescribed extent of oxygen change and N₂O-SP and N₂O-$\delta^{18}$O values, all ranges suggest that biotic denitrification instead of chemodenitrification was the main contributor to N₂O production in the plastisphere and bulk water.

by b*nirK*-type denitrifiers. In light of the Bray-Curtis distances (Fig. 6a), the three denitrifier communities between plastisphere and bulk water clearly formed two separated clusters ($P < 0.01$, $R = 0.873–0.927$). No obvious or separated clusters among the four plastic debris were observed. As shown in Supplementary Fig. 7, the major phylum was Proteobacteria in b*nirS*-type and b*nirK*-type denitrifier communities, and it was Ascomycota and Basidiomycota in f*nirK*-type denitrifier communities. However, their relative abundances varied greatly between plastisphere and bulk water, implying a distinct denitrifier structure between these compartments. We also found that plastic types did not obviously change the denitrifier compositions and relative abundances (Supplementary Fig. 7), albeit with some variations in b*nirK*-type denitrifier community at the genus level for the polystyrene (PS) type.

Co-occurrence network analysis showed that the plastisphere harbored more complex interactions among the three denitrifier communities in comparison with bulk water (Fig. 6b and Supplementary Table 1). We found that the keystone genera of denitrifiers were shifted between plastisphere and bulk water. In the plastisphere, the b*nirK*-type denitrifiers (*Pseudomonas*, *Bosea*, *Ochrobactrum*, and *Starkeya*) accounted for the main keystone genera, while b*nirS*-type denitrifiers (α,β-Proteobacteria and *Azoarcus*) were dominant in bulk water (Fig. 6b). f*nirK*-type keystone denitrifiers were also changed from *Talaromyces*, *Chaetomium*, and *Penicillium* in the plastisphere to *Fusarium*, *Alternaria* and *Trichosporon* in bulk water. Such shifts probably explained the different denitrifying activities between plastisphere and bulk water. In addition, we also quantified the relative abundances of *narG*, *nirS*, *nirK*, and *nosZ* (Supplementary Fig. 8). Plastisphere generally contained higher abundances than the bulk water ($P = 0.001–0.038$), partially explaining the higher denitrification rates in the plastisphere.

We further explored the denitrifier-based niche breadth and overlap at community level. Niche breadths of all three denitrifier communities in the plastisphere were greater than those of bulk water ($P < 0.001$, Fig. 6c). This indicates that plastic surfaces could offer more opportunities for denitrifiers attachment and biofilm development than the surrounding water[46]. In line with the results of niche breadth, the plastisphere also presented a higher niche overlap level ($P < 0.001$, Fig. 6d), which reflects a sharp competition in denitrifier communities for energy and nutrients in comparison with bulk water[47].

## Discussion

The potential of plastisphere in global nitrogen cycling and associated greenhouse gas flux is rarely examined. Our study provides evidence of the plastisphere exhibiting a higher denitrifying activity compared to bulk water and functioning as a neglected source of greenhouse gas N₂O production. Below, we discuss the likely reasons for these findings.

In our study, higher levels of PS, PN, and eDNA were observed in the plastisphere comparing with bulk water (Fig. 2b–d). This indicates that the estuarine microbes secrete EPS and then grow within the self-produced extracellular matrix to form biofilms during their colonization on plastic debris[24,48]. We also found that the EPS concentrations were coordinated with the c-di-GMP levels that were higher in the plastisphere and lower in bulk water (Supplementary Fig. 3). This is likely because microbial secretion of EPS and biofilm formation are mainly regulated by the intracellular c-di-GMP level, especially in the Proteobacteria phylum[25,43]. Recent studies have demonstrated that higher c-di-GMP levels could enhance EPS yields through repressing the expression of transcription factor FleQ (*P. aeruginosa*)[49] or increasing the surface adherence and immobility organelles such as T4P (*V. cholerae* and *C. difficile*)[50]. In addition, c-di-GMP is also capable of downregulating flagellar expression (*P. aeruginosa*), reducing flagellar assemblies (*C. crescentus*)[43], or perturbing flagellar motor function (*E. coli*)[51], which suppresses the motility and gradually forms biofilms. Therefore, the elevated c-di-GMP levels in the plastisphere are important for EPS production and biofilm formation on the plastic debris.

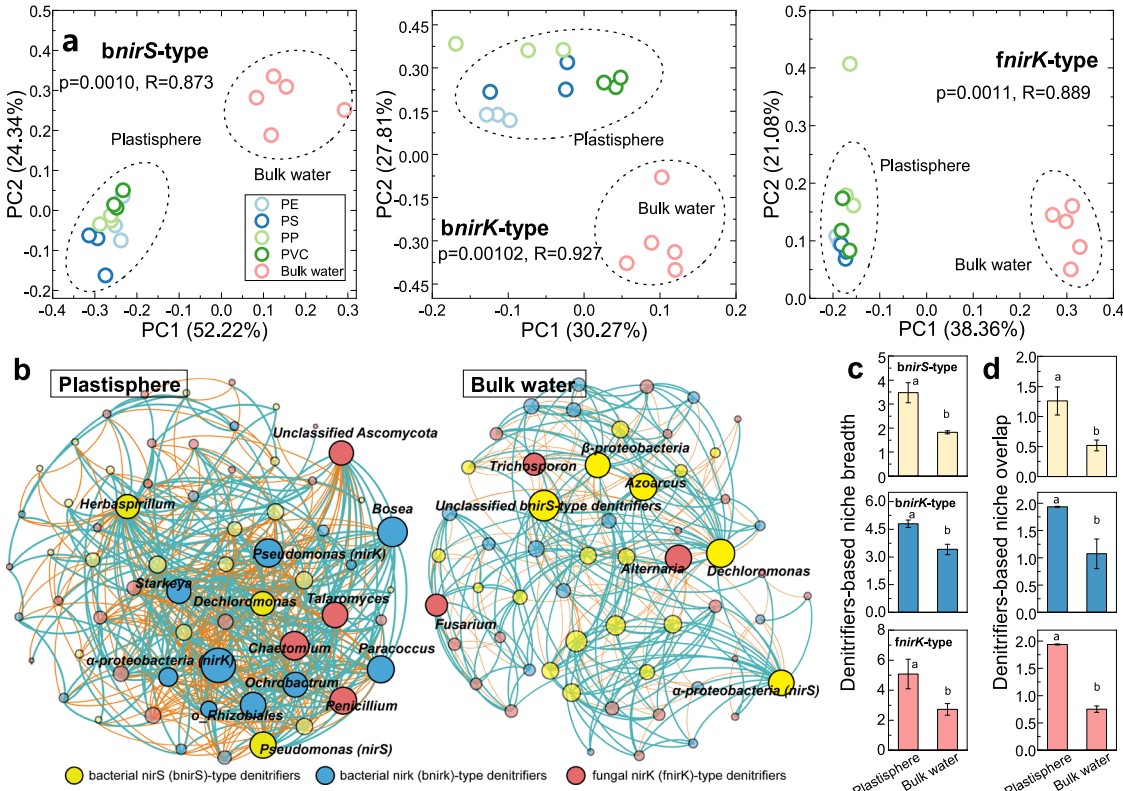

**Fig. 6 Denitrifier community in the plastisphere and bulk water. a** Principal coordinates analysis with ANOSIM test showing the differences in community structures of b*nirS*-type, b*nirK*-type, and f*nirK*-type denitrifiers between plastisphere and bulk water. **b** Co-occurrence network analysis illustrating the keystone denitrifiers (nodes, $R^2 > 0.9$, $P < 0.05$) in the plastisphere and bulk water. The parameters of the networks are listed in Supplementary Table 1. b*nirK*-type denitrifiers dominated in the keystone taxa in the plastisphere, while b*nirS*-type ones were the main keystone in bulk water. **c**, **d** Niche breadth and overlap based on the three types of denitrifier communities. Error bars are the standard error. Different letters (a and b) indicate the significant differences ($P < 0.05$) between plastisphere and bulk water.

Another important finding is that microbial lipid and fatty acid levels were lower in the plastisphere than in bulk water (Fig. 3). Intracellular lipid/fatty acid signals are reported to be involved in biofilm formation and dispersion[44]. Accumulations of lipid/fatty acids in microorganisms could activate the activity of phospho-diesterases (catalyzing c-di-GMP consumption pathway), leading to lower c-di-GMP levels and biofilm dispersion[25]. Moreover, lipid/fatty acid signaling molecules, such as cis-2-decenoic acid, cis-11-methyl-2-dodecenoic acid, and cis-2-dodecenoic acid, are capable of dismantling small microaggregates and inhibiting biofilm formation[52,53]. For example, previous studies have shown that biofilm roughness, thickness, and biomass were significantly reduced in the presence of cis-2-decenoic acid[52]. This is likely because these signaling molecules could maintain bacterial and fungal cells metabolically active, benefit for dispersive growth. Thus, the lower intracellular lipid/fatty acid levels observed here might be another important reason for the biofilm formation on the plastics.

During biofilm formation, surfaces of plastics are more likely to recruit anoxic denitrifiers, which is substantiated by the higher abundances of denitrifiers and denitrifying genes in the plastisphere than in bulk water (Supplementary Figs. 7 and 8). Low concentrations of dissolved oxygen in biofilms may account for the result. In biofilms, a steep gradient of dissolved oxygen could occur from outer to inner spaces, and thus produce anoxic conditions[54,55]. Since denitrifiers can utilize nitrate/nitrite as electron acceptors instead of $O_2$ to support their metabolisms[20], the oxygen-limited conditions are favorable for their growth and activity. The higher anoxic activities in the plastisphere (Fig. 1)

further supported this explanation. In addition, we also found that the bacterial denitrifiers in the plastisphere mainly belonged to Proteobacteria phylum (Supplementary Fig. 7), consistently with previous studies indicating that Proteobacteria was the globally most abundant on surfaces of marine plastics[56]. Many members of Proteobacteria (such as *Pseudomonas*) display strong capacities of EPS secretion and have evolved a "holdfast"[57]. Thus, these denitrifiers may be favored compared to other microorganisms for surface attachment onto plastic debris. Taken together, our study revealed that, on contact with the plastic debris, estuarine microorganisms produced EPS to form biofilm by the regulation of c-di-GMP and lipid/fatty acid signals. The subsequently generated anoxic microsites could provide opportunities for denitrifiers, and thereby resulted in the higher denitrifying activity in the plastisphere, supporting our first hypothesis.

Denitrification and $N_2O$ production were recorded in the plastisphere and bulk water (Fig. 4). The observed denitrification rates are of comparable magnitude to previous studies conducted in remote oceans[58], coastal, and estuarine sediments[21], suggesting the importance of the denitrification process in marine ecosystems. Comparing with bulk water, plastisphere had greater denitrification rates and $N_2O$ productions (Fig. 4), which supports our second hypothesis. It can be soundly attributed to the differences of denitrifier communities and biomasses between plastisphere and bulk water. Our study showed that the niche breadths of denitrifier communities in the plastisphere were higher than that of bulk water (Fig. 6c), implying that more denitrifiers are likely to be attached to the plastics than to freely

live in water[14,46]. Network analysis further indicated that b$nirK$-type denitrifiers dominated in the keystone taxa of plastisphere, whereas b$nirS$- type ones were the core residents in bulk water (Fig. 6b). In comparison with b$nirS$-type denitrifiers, b$nirK$-type ones could display strong adaptability to fluctuating environments (i.e., dissolved oxygen)[59], and thus are more easily attached onto the plastics. Furthermore, b$nirK$-type denitrifiers are more effective for nitrite reduction and $N_2O$ production than b$nirS$-type ones[60]. This is further evidenced by the greater ratios of ($nirS + nirK$)/$nosZ$ (1.5–2.3) in the plastisphere than those (0.33–0.41) in bulk water (Supplementary Fig. 8). All of these accounted for the stronger denitrifying capability in the plastisphere.

Using the mass balance model, we found that BD dominated in $N_2O$ production in both the plastisphere and bulk water (Fig. 5b). Generally, the oxygen-limited conditions are more favorable for BD process. The bacterial denitrifiers (b$nirS$- or b$nirK$- type), identified as the main keystone taxa in the plastisphere and bulk water (Fig. 6b), further supported the bacterial dominance in $N_2O$ productions. The bacterial proportions in this study (51–79%) were higher than those of previous studies conducted in coastal (14–60%)[28] and estuarine (15–21%)[61] sediments, suggesting a spatial heterogeneity of BD in marine ecosystems. In addition, we noticed that the bacterial contributions to $N_2O$ production had no striking difference among the four types of plastics (Fig. 5b), which signifies the intrinsic importance of BD, rather than an opportunistic response in the estuarine plastisphere. The enhanced contribution of BD is favorable for reactive nitrogen removal by the transformation to $N_2O$ and $N_2$. This implies that BD is an important source for greenhouse gas $N_2O$.

Fungal contributions to $N_2O$ production in estuarine water have been less well studied. It generally derives from two pathways including FD and fungi-mediated codenitrification[31]. During code-nitrification with the addition of $^{15}N$-nitrate/nitrite, fungi would form more hybrid $^{45}N_2O$ and $^{29}N_2$ than $^{46}N_2O$ and $^{30}N_2$[62,63]. In our $^{15}N$-labeling experiment, however, we detected a large portion of $^{30}N_2$ rather than $^{29}N_2$, indicating that the codenitrification contribution to $N_2O$ production was less in this study. Fungal $N_2O$ production could thus be mainly from FD process. Our study further showed that the contributions of FD to total $N_2O$ productions were 8–46% in the plastisphere and bulk water (Fig. 5b). This highlighted the non-negligible role of fungal denitrifiers in $N_2O$ production, despite the proportions being lower than those in marine sediments (40–70%)[28], which partially supports our third hypothesis. Moreover, we found higher proportions of $N_2O$ production from FD in the plastisphere (26–46%) than in bulk water (8–22%, Fig. 5b). Fungal niche selection may account for the different $N_2O$ production patterns between plastisphere and bulk water, because the aquatic microorganisms are prone to nesting in more fragmented habitats like plastic debris rather than in water[14]. The biofilms formed by attached microbes promote cellular interactions, thus likely leading to the distinct $N_2O$ dynamics between plastisphere and bulk water.

The small contribution of CD to $N_2O$ production (0–10.28%) indicated that CD is not an important $N_2O$ contributor in both the plastisphere and bulk water (Fig. 5b). By contrast, Wankel et al.[28] and Wang et al.[38] documented that the contributions of CD to $N_2O$ production could reach 31–75% in paddy soils and coastal sediments. The different results were probably caused by the different iron contents that play an essential role in the CD process[64]. In soils and sediments, the chemical reactions of particle-bound reduced iron with various enzymatically-derived electron acceptors (i.e., hydroxylamine, nitric oxide or nitrite) can contribute to pronounced yields of $N_2O$[64]. In our study, however, the iron concentrations were low (2.89–7.95 mg $L^{-1}$), thus likely limiting the importance of the CD process in estuarine water.

It is worth noting that the nitrification process in the plastisphere and bulk water, including hydroxylamine oxidation and nitrifier denitrification[41,65], may also contribute to $N_2O$ production. We further conducted a lab-scale experiment under in situ dissolved oxygen concentration (~1.2 mg $L^{-1}$, hypoxic conditions) to explore nitrifier potential in $N_2O$ production in the plastisphere and bulk water (Supplementary Fig. 9). There was no significant difference observed between groups with and without adding allylthiourea (a nitrification inhibitor[66]), indicating that $N_2O$ production via hydroxylamine oxidation was minimal under in situ conditions. By contrast, nitrifier denitrification was likely a potential contributor in the plastisphere and bulk water (Supplementary Fig. 9b), but the concentration was lower than that of denitrification-based $N_2O$. More importantly, the nitrifier denitrification-based $N_2O$ production in the plastisphere was higher than in bulk water (Supplementary Fig. 9). This further supports that the plastisphere is an overlooked source of $N_2O$ production. Although in-depth discussions on the nitrifier-related sources of $N_2O$ are beyond the scope of this study, we believe that it is urgent to investigate nitrification and associated $N_2O$ dynamics in future research, which will deepen the understanding of the role of aquatic plastisphere in biogeochemical cycling.

Plastisphere derived from the widespread dissemination of plastic debris in estuarine environments[8], is changing the stability and function of the ecosystem[12,14]. Our results reveal that the plastisphere produces approximately twofold greater emissions of the potent greenhouse gas $N_2O$ from denitrification pathways than the bulk water. By further assessing Global Warming Potential (GWP) of $N_2O$ production[67], we also find that plastisphere exhibits the higher warming potential (34.86–41.12 mg $CO_2$ eq $L^{-1}$ $d^{-1}$) than the bulk water (13.15–17.26 mg $CO_2$ eq $L^{-1}$ $d^{-1}$). Both of these indicate that the emerging plastisphere could be a great but neglected source of anthropogenic $N_2O$. Although our data could not make precise estimations of $N_2O$ flux and its warming potential from global plastisphere, the importance of estuarine plastisphere in $N_2O$ production via anthropogenic sources should be recognized, especially under global climate change. It is estimated that annual emissions of plastic debris to marine ecosystems are 5–13 million tons[8], and the biomass on these plastics could be 0.2–6% of total microbial biomass in marine surface waters[10]. Given the substantial plastic waste and microbial biomass, future research at larger scales is needed to make reliable predictions of $N_2O$ fluxes from plastisphere and to evaluate the warming potential of $N_2O$ production from pervasive plastisphere across global marine ecosystems.

## Methods

**Study site, plastic preparation, and experimental design.** In this study, the in situ incubations were conducted in an estuary of Xiamen, China (118°07'E, 24°61'N–118°09'E, 24°59'N) during April and May 2021. This region possesses a subtropical maritime monsoon climate, and has an average temperature of 21 °C and receives 1100 mm of rain[68]. Owing to human activities, pollutants in rainwater runoff and a portion of wastewater have been recently transported to the estuary, leading to reduced water quality and slight eutrophication. We sampled the estuarine water at 4 m depth and determined its characteristics. The water (pH 6.5, 20 °C) contained 1.19 mg $L^{-1}$ dissolved oxygen, 21.2–25.7 mg $L^{-1}$ total organic carbon, 0.72~0.99 mg $L^{-1}$ nitrate, 0.19–0.43 mg $L^{-1}$ ammonium, 0.09~0.17 mg $L^{-1}$ nitrite, and 2.89~7.95 mg $L^{-1}$ iron.

Four types of common commercial plastics, i.e., polyethylene (PE), polystyrene (PS), polypropylene (PP), and polyvinylchloride (PVC), were selected. They have a low bioreactivity with densities of 0.88–0.97 g cm$^{-3}$[31]. The PE-type food bags were from Cleanwrap Co., China. PS was obtained from transparent plastic cups (Chuanguan Co., China). PP was from sealable plastic bags from Xingmin Co., China. The PVC-type cling films were donated by Jusu Co., China. The four types of plastics were cut into a size of ~10 × 10 cm for subsequent experiments.

To investigate the denitrifying capability of the plastisphere, we performed (1) an in situ incubation for 30 days (30d), and (2) a subsequent denitrification experiment for 24 h (24 h). The detailed experimental design is illustrated in Supplementary Fig. 1. At the initial stage of in situ incubation, the obtained 10 × 10-cm plastic debris was sterilized using 70% ethanol and connected with

cotton cords, and we used stone debris placed under the plastics to maintain buoyancy. This led to a ~4 m water depth for the plastics suspended in estuarine water. After 30 days, the plastic debris and surrounding bulk water (10 L) were harvested with a polymethyl-methacrylate sampler (Haifuda, China). A portion of plastics and bulk water was used for water quality analysis, SEM, and D$_2$O-labeled single-cell Raman spectroscopy. The remaining material was used for the preparation of plastisphere suspensions and the subsequent 24 h experiment. Methods for water quality detections and SEM are listed in Supplementary Methods 1 and 2.

Prior to the denitrification experiment, a portion of bulk water was first filtered through a 0.22-μm polycarbonate membrane to remove microorganisms and then transferred into 250-mL sterile beakers. Thereafter, the collected plastic debris was placed in these beakers, and biofilms were repeatedly shaken, agitated, and washed with the sterile bulk water until a large proportion of the biofilms were detached[69]. Plastisphere suspensions were then collected for the determinations of denitrification rate, N$_2$O reduction ratio (f$_R$), EPS and c-di-GMP levels, and further experiments (Supplementary Fig. 1). During the 24 h denitrification experiment, we established plastisphere groups (i.e., plastisphere suspensions of PE, PS, PP, and PVC) and a bulk water group (estuarine water). Each plastisphere group was presented in triplicate, and the bulk water group (in quintuplicate) was set as control. Forty milliliters of plastisphere suspensions and bulk water were transferred into 120-mL serum bottles, respectively. Next, these bottles were sealed, purged with high-purity He (>99.99%) for 5 min, and incubated at 20 °C for 24 h. No additional carbon or nitrogen source was added, aiming to explore the basal capacities of denitrification in the plastisphere and bulk water. Each lab-scale experiment was conducted in parallel, and variations in the concentrations of organic matter during the lab-scale experiment are shown in Supplementary Fig. 10. After 24 h, N$_2$O concentration and isotopocules, denitrifier community, and gene abundance in the plastisphere and bulk water were measured.

In this study, we prepared the plastisphere suspensions, aiming to explore the biofilms on the plastics under more controlled conditions during the lab-scale experiments. To examine if the growth pattern (attached or detached biofilms) has significant impacts on microorganisms during the 24 h experiment, we further investigated N$_2$O production, anoxic microbial activities, and community structure of the attached and detached biofilms (Supplementary Fig. 11). These results showed no significant differences between attached and detached biofilms, suggesting that denitrifier activities during 24 h were not likely to be significantly impacted by growth patterns. In addition, we selected the water volume of 40 mL to conduct the 24 h experiment, because it is the minimum volume for a large proportion of biofilm detachment from plastic debris. We also conducted the same experiments with different volumes (80 and 120 mL) to avoid bias induced by incubation water volume (Supplementary Fig. 12). The results showed that N$_2$O production in the plastisphere was higher than that in bulk water, regardless of water volumes.

In addition, it should be noted that marine particle debris is mainly in the form of plastics, occupying up to 60–80%[70]. Moreover, other particle fragments such as glass, metal, or stone, could be deposited to marine sediments, and the direct effects of these particles on marine waters are minor. Therefore, we only consider the plastic debris in this study.

**EPS and c-di-GMP detections**. After the 30-d incubation, the EPS and c-di-GMP concentrations in the plastisphere and bulk water were measured. Extractions of EPS were following the formaldehyde-NaOH method[71]. Plastisphere suspension (3 mL) or bulk water (3 mL) together with 2 mL sterile phosphate buffer were transferred into a 15-mL centrifuge tube. After centrifugation (5000×g, 4 °C, 5 min) and removal of the supernatant, 5 mL of sterile phosphate buffer and 50 μL of 37% formaldehyde were added to re-suspend the pellet. The tube was shaken for 5 min and stored at 4 °C for 60 min, and then 2 mL of 1 M NaOH was added. After being stored at 4 °C for 3 h, the supernatant was harvested by centrifugation (5000×g, 4 °C, 10 min) and used for PS, PN, and eDNA detections after filtering through a 0.22-μm polycarbonate membrane. PS concentration was measured at $\lambda = 625$ nm following the Anthrone method using glucose as standard. PN concentration was determined at $\lambda = 562$ nm by a Protein Assay Kit (Beyotime Biotechnology, China) using BSA as standard. eDNA concentration was quantified at $\lambda = 260$ nm by a DNA Detection Kit (QuantiFluor, dsDNA system, USA).

Microbial c-di-GMP in the plastisphere and bulk water were extracted and measured based on previous studies[72,73]. The extraction solvent consisted of acetonitrile/methanol (50/50, v/v). Cells in the plastisphere suspension and bulk water were disrupted using ultra-sonication for 5 min (120 W, 4 °C), and then transferred into a 15 mL centrifuge tube. After adding 8 mL extraction solvent and vortexing for 60 s, the suspensions were maintained at −80 °C for 2 h, and the supernatants were collected by centrifugation (14,000×g, 15 min). Thereafter, a pressured gas blowing concentrator (HGC-12A. Hengao, China) was used to dry the supernatants under continuous flows of N$_2$. The residues were resuspended in 500 μL of ultrapure water for subsequent detection of c-di-GMP. The concentration of c-di-GMP was detected with an LC-MS/MS equipped with an electrospray ionization source (Applied Biosystems 6500, USA) at a reversed-phase. The m/z + 689/150 and + 689/343 quantifiers were c-di-GMP transitions. Declustering potential (DP) was −185 and collision energy (CE) was −45 and −48. The obtained peak areas were compared with a calibration curve generated with a c-di-

GMP standard (Sigma-Aldrich, Germany). Concentrations of intracellular proteins were measured using a Protein Assay Kit (Beyotime Biotechnology, China).

**Denitrification rate and N$_2$O isotope measurement**. After the 30-d incubation, denitrification rates of the plastisphere and bulk water were measured using the $^{15}$N isotope-pairing technique[21,74]. Five milliliters of plastisphere suspension or bulk water were prepared and transferred to a 12 mL gas-tight vial (Labco Exetainer, UK). The vials were purged with high-purity He (99.99%) for 5 min to replace the headspace air. The samples were first pre-incubated at 20 °C for 24 h to remove residual nitrate. After the removal, the vials were spiked with K$^{15}$NO$_3$ solution (99.8% atom, Macklin Co. China) to reach a concentration of 200 μM $^{15}$NO$_3^-$ on the basis of in situ concentrations, and then re-purged with high-purity He. The samples were incubated in dark for 8 h at 20 °C. After the incubation, 2 mL of headspace gas was taken by a syringe and transferred to a new 12 mL vial pre-filled with He gas. The concentrations of $^{28}$N$_2$, $^{29}$N$_2$, and $^{30}$N$_2$ in the headspace were quantified with a GasBench-II/isotope ratio mass spectrometry (Thermal Delta V Advantage, Germany). The detection limit is 0.1 μM. The denitrification rate was calculated as follows:

$$\text{Denitrification rate} \, (\text{mg}^{15} N \, L^{-1} h^{-1}) = \frac{T_{30} \times 2 \times (1 - F_n)}{F_n} + 2 \times T_{30} \quad (1)$$

where $T_{30}$ (mg $^{30}$N$_2$ L$^{-1}$ h$^{-1}$) represents the production rates of $^{30}$N$_2$ during the 8 h. $F_n$ (99%) is the $^{15}$N fraction in added $^{15}$NO$_3^-$ to residual $^{15}$NO$_3^-$ after pre-incubation.

After the 24-h denitrification experiment, the headspace gas from the 120-mL serum bottle was taken to measure N$_2$O concentrations and isotopic compositions. Three milliliters of headspace gas was used to analyze N$_2$O concentration with a gas chromatograph (7890A, Agilent Technologies, USA), and 1 mL of headspace gas was used to detect N$_2$O isotope compositions, including δ$^{15}$N$^{bulk}$, δ$^{15}$N$^{\alpha}$, δ$^{15}$N$^{\beta}$, and δ$^{18}$O, with a Precon+Gasbench/isotope ratio mass spectrometer (Delta V plus, Thermal, USA)[75]. Of these, $^{15}$N$^{\alpha}$ (central, $^{14}$N–$^{15}$N–$^{16}$O) and $^{15}$N$^{\beta}$ (terminal, $^{15}$N–$^{14}$N–$^{16}$O) are the isotopocules[76]. N$_2$O gas in samples was first enriched in liquid N$_2$ trap, purified, and then separated by a gas chromatography column (30-m long, 0.32-mm i.d.). Next, high-purity He flowing at 2 mL min$^{-1}$ was applied to transport the N$_2$O gas to the mass spectrometer. N$_2$O isotopocules were detected by simultaneously capturing NO$^+$ (m/z 30, 31) and N$_2$O$^+$ (m/z 44, 45, 46). The scrambling factor was determined as 0.085. In this study, the isotopocule values of the internal reference gas (high-purity N$_2$O (>99.99%)) were pre-analyzed in Thünen Institute of Climate-Smart Agriculture, Germany. Two standard gases provided by Dr. Anette Goeske and Dr. Reinhard Well were applied to perform two-point calibrations for values of N$_2$O site preference (N$_2$O-SP, AK1 N$_2$O-SP = −2‰, Mix 1 N$_2$O-SP = 15‰, 5ppm). Values of δ$^{15}$N$^{bulk}$, δ$^{15}$N$^{\alpha}$, δ$^{15}$N$^{\beta}$, δ$^{18}$O, and SP of N$_2$O were calculated as follow:[77,78]

$$\text{N}_2\text{O} - \delta^{15}\text{N}^i(‰) = (^{15}\text{N}_{sample}^i - ^{15}\text{N}_{standard}) / ^{15}\text{N}_{standard} (i = \alpha \, \text{or} \, \beta) \quad (2)$$

$$\text{N}_2\text{O} - \delta^{18}\text{O}(‰) = (^{18}\text{O}_{sample} - ^{18}\text{O}_{standard}) / ^{18}\text{O}_{standard} \quad (3)$$

$$\text{N}_2\text{O} - \delta^{15}\text{N}^{bulk}(‰) = (\delta^{15}\text{N}^{\alpha} + \delta^{15}\text{N}^{\beta}) / 2 \quad (4)$$

$$\text{N}_2\text{O} - \text{SP}(‰) = \delta^{15}\text{N}^{\alpha} - \delta^{15}\text{N}^{\beta} \quad (5)$$

where δ$^{18}$O and δ$^{15}$N$^i$ are the ratios of $^{18}$O/$^{16}$O and $^{15}$N/$^{14}$N, respectively. δ$^{15}$N$^{\alpha}$ and δ$^{15}$N$^{\beta}$ denote the ratios of $^{14}$N–$^{15}$N–$^{16}$O and $^{15}$N–$^{14}$N–$^{16}$O, respectively. All values were presented as ‰ relative to atmospheric N$_2$ ($^{15}$N) or Vienna Standard Mean Ocean Water ($^{18}$O). The typical detection precisions are 0.3‰ (N$_2$O-δ$^{15}$N$^{bulk}$), 0.9‰ (N$_2$O-δ$^{15}$N$^{\alpha}$), 0.9‰ (N$_2$O-δ$^{15}$N$^{\beta}$) and 0.6‰ (N$_2$O-δ$^{18}$O).

**Correction of N$_2$O-o$^{18}$O and N$_2$O isotope mass balance model**. During denitrification, the incorporation of the oxygen atom from H$_2$O into N$_2$O or nitrate could affect the measured δ$^{18}$O values of N$_2$O (N$_2$O-δ$^{18}$O)[75,79,80]. Previous studies conducted in soils suggest that an almost oxygen exchange occurred during denitrification[79]. Therefore, assuming a complete O exchange δ$^{18}$O values of the ambient water (H$_2$O-δ$^{18}$O) could be used for interpretation. As the variation in H$_2$O-δ$^{18}$O values in global surface seawater is minor[81], we used a theoretical H$_2$O-δ$^{18}$O value (0.91‰, between −2.98 and 1‰)[81,82] reported in the literature for seawater to correct the measured N$_2$O-δ$^{18}$O values (subtracting H$_2$O-δ$^{18}$O value).

After correction, the multiple-isotope relationships between N$_2$O-SP and N$_2$O-δ$^{18}$O were plotted based on the mapping approach[83], which was first applied by Lewicka-Szczebak et al. Then, we further established a N$_2$O isotope mass balance[28] to explore the relative fractions of N$_2$O-producing sources in the plastisphere and bulk water. As the denitrification experiment was anaerobically executed, N$_2$O productions via BD, FD, and CD processes were taken into account. Fractions of BD (f$_{BD}$), FD (f$_{FD}$), and CD (f$_{CD}$) pathways to total N$_2$O flux were expressed as follow:

$$\text{Total N}_2\text{O production} = f_{BD} + f_{FD} + f_{CD} = 1 \quad (6)$$

For N$_2$O-SP values, N$_2$O isotope mass contributions of BD, FD, and CD processes

to the total $N_2O\text{-}SP_{NR}$ values production were expressed as:

$$N_2O - SP_{NR} = f_{BD} \times N_2O - SP_{BD} + f_{FD} \times N_2O - SP_{FD} + f_{CD} \times N_2O - SP_{CD} \tag{7}$$

where $N_2O\text{-}SP_{NR}$ refers to the $N_2O\text{-}SP$ value before reduction. $N_2O\text{-}SP_{BD}$, $N_2O\text{-}SP_{FD}$, and $N_2O\text{-}SP_{CD}$ are the $N_2O\text{-}SP$ values of BD ($-7.5$–$3.7$‰)[65,76,84], FD ($27.2$–$39.9$‰)[75,76,85–87], and CD ($10$–$22$‰)[29,32,76,88], respectively. Similarly, the contributions of the three endmembers to $N_2O\text{-}\delta^{18}O$ values of the three processes were expressed as:

$$N_2O - \delta^{18}O_{NR} = f_{BD} \times N_2O - \delta^{18}O_{BD} + f_{FD} \times N_2O - \delta^{18}O_{FD} + f_{CD} \times N_2O - \delta^{18}O_{CD} \tag{8}$$

where $N_2O\text{-}\delta^{18}O_{NR}$ refers to the $N_2O\text{-}\delta^{18}O$ value before reduction. $N_2O\text{-}\delta^{18}O_{BD}$, $N_2O\text{-}\delta^{18}O_{FD}$, and $N_2O\text{-}\delta^{18}O_{CD}$ are the $N_2O\text{-}\delta^{18}O$ values of BD ($16.4$–$23.3$‰)[65,76,84], FD ($42.0$–$55.1$‰)[75,76,85–87], and CD ($24.9$–$37.6$‰)[29,32,76,88], respectively.

Nitrogen and oxygen isotope fractionations during $N_2O$ reduction such as BD can increase the measured $N_2O\text{-}SP$ and $N_2O\text{-}\delta^{18}O$ values of remaining $N_2O$. Fractionation-derived shifts in $N_2O\text{-}SP$ and $N_2O\text{-}\delta^{18}O$ values during $N_2O$ reduction were thus estimated according to a previous study and presented as:

$$N_2O - SP = N_2O - SP_{NR} - f_R \times SP_{eN2O} \tag{9}$$

$$N_2O - \delta^{18}O = N_2O - \delta^{18}O_{NR} - f_R \times \delta^{18}O_{eN2O} \tag{10}$$

where SP and $\delta^{18}O$ are the $N_2O\text{-}SP$ and $N_2O\text{-}\delta^{18}O$ values after $N_2O$ reduction, respectively, which are the measured SP and $\delta^{18}O$ values here; $f_R$ denotes the $N_2O$ reduction ratio (0.75 in this study). This was estimated using the $C_2H_2$ approach[41], i.e., the proportion of the residual (without $C_2H_2$; residual $N_2O$ after $N_2O$ reduction) to total (with $C_2H_2$; residual $N_2O$ after $N_2O$ reduction) $N_2O$ concentrations. $SP_{eN2O}$ ($-6$‰) and $\delta^{18}O_{eN2O}$ ($-25$‰) were assumed to be the fractionation effect during $N_2O$ reduction[76,89]. Given the $N_2O$ reduction effect during experiments, Eqs. (7) and (8) were modified as:

$$N_2O - SP = f_{BD} \times N_2O - SP_{BD} + f_{FD} \times N_2O - SP_{FD} + f_{CD} \times N_2O - SP_{CD} - f_R \times SP_{eN2O} \tag{11}$$

$$N_2O - \delta^{18}O = f_{BD} \times N_2O - \delta^{18}O_{BD} + f_{FD} \times N_2O - \delta^{18}O_{FD} + f_{CD} \times N_2O - \delta^{18}O_{CD} - f_R \times N_2O - \delta^{18}O_{eN2O} \tag{12}$$

We further applied the Gaussian estimation and Monte Carlo sampling method to obtain the mean values and standard errors of $N_2O\text{-}SP$ and $N_2O\text{-}\delta^{18}O$ for each endmember, and assess the errors of the mass balance model. The detailed MATLAB codes and Monte Carlo stimulations (10,000 sampling) are provided in Supplementary Table 2 and Supplementary Fig. 13. The values of SP and $\delta^{18}O\text{-}(N_2O/H_2O)$ used in the model for BD were $-1.5$‰ and 19‰, respectively; for FD were 37‰ and 47‰; for CD were 16‰ and 30‰. Combining Eqs. (6), (11), and (12), the relative contributions of BD, FD, and CD to $N_2O$ productions are acquired in the plastisphere and bulk water. Proportions estimated by the equations are considered as zero if yielding negative values, and the other two proportions sum up to 100% accordingly.

**D$_2$O-labeled single-cell Raman spectroscopy**. To compare microbial phenotypic activity and to explore intracellular lipid/fatty acid levels between plastisphere and bulk water, a $D_2O$-labeled single-cell Raman spectroscopy experiment was performed[15,90]. After the 30-d incubation, the four types of plastics were cut to produce ~$1.5 \times 1.5$ cm plastic fragments. For the plastisphere, 1 mL of sterilized bulk water and the plastic fragments were transferred to a 12-well plate and mixed with 1 mL $D_2O$, finally reaching a concentration of 50% $D_2O$. For bulk water, 1 mL of in situ bulk water and 1 mL $D_2O$ were added to another 12-well plate. Each group was conducted in triplicate. The 12-well plates were anaerobically incubated at 300 rpm and 20 °C for 24 h. Next, plastic fragments were transferred to 5 mL phosphate buffer with 0.5% Tween-20, and attached microorganisms were separated by vortexing for 10 min. After washing with sterilized water twice (4000×g, 5 min), 2 μL of microorganisms from bulk water and plastisphere were spotted on a tin foil-coated microslide and dried at 25 °C. Raman spectroscopy was acquired with a LabRAM Aramis (HORIBA Jobin-Yvon) confocal micro-Raman system, including a 300-g/mm diffraction grating, a 532-nm Nd:YAG laser, and a ×100 objective (Olympus, Japan). We used the Raman characteristic band of a silicon wafer ($520.6\ cm^{-1}$) to calibrate the Raman spectrometer. The generated spectra were subsequently analyzed in LabSpec-5 software (HORIBA Jobin-Yvon) with baseline correction and normalization. Peak intensities of Raman bands assigned to C–D ($2040$–$2300\ cm^{-1}$) and C–H ($2800$–$3100\ cm^{-1}$) were obtained to estimate (C–D)/(C–D + C–H) ratios[42].

**Denitrifier community**. After the 24 h denitrification experiment, 2 mL of plastisphere suspension was taken for DNA extraction with a FastDNA Kit (MP, SA, USA). In situ bulk water samples were first filtered through sterilized 0.22μm polycarbonate filters that were then used for DNA extraction[15]. Bacterial nirS gene (bnirS-type denitrifier), bacterial nirK gene (bnirK-type denitrifier), and Internally

Transcribed Spacer (ITS) gene (fungal community) were used for amplicon sequencing with the primer sets (Supplementary Table 3). The obtained raw data were quality-filtered, chimera checked and then grouped at 97% similarity. Taxonomy was compared with GeneBank and UNITE databases using QIIME. Information of fungal nirK denitrifiers (fnirK-type denitrifiers) was screened from fungal communities based on previous studies and NR database[36,91]. The coverage of screened operational taxonomic unit (OTU) to total OTUs of ITS data was >85%, which was used for subsequent analysis. The α-diversities including abundance-based coverage estimation (ACE), Chao1, Shannon, and Simpson indexes, were estimated in R. Principal coordinates analysis coupled with analysis of similarities (ANOSIM) test was applied to explore β-diversities of the three types of denitrifier communities. Co-occurrence network analysis was performed in R[92] (nodes $R^2 > 0.90$ and $P < 0.05$) and visualized by Gephi (Version 0.9.2). Sequences will be submitted under accession number SUB10278422.

We investigated denitrifiers-based niche breadth and overlap according to Levins' niche breadth method and asymmetric α-model in R[14,46,47]. These two indicators reflect the potential of denitrifiers adapting to environments and competing for resources in the plastisphere and bulk water. The formulas were as follow:

$$B_i = \frac{1}{\sum_{i=1}^{N} \times P_{ij}^2} \tag{13}$$

and

$$\alpha_{ij} = \frac{\sum_{i=1}^{N} \times P_{i\alpha} \times P_{j\alpha}}{\sum_{i=1}^{N} \times P_{i\alpha}^2} \tag{14}$$

where $B_i$ and $\alpha_{ij}$ are the niche breadth and overlap of denitrifier i in the plastisphere or bulk water, respectively; $N$ denotes the total number of denitrifiers in each type of denitrifier community; $P_{ij}$ denotes the proportion of denitrifier i in communities.

**Gene abundance**. The abundances of bacteria (16S rRNA) and fungi (ITS) in the plastisphere and bulk water were quantified with qPCR using the universal primers 515F/907R and ITS1F/ITS2R, respectively[68,92]. The standard curves were obtained by a serial dilution of the plasmids with 16S rRNA and ITS genes. The mixing system (20 μL) included 2 μL of DNA sample, 10 μL of Mix-enzyme (LightCycler SYBR Green I), 0.8 μL of 515 F/907 R or ITS1F/ITS2R primers, and 7.2 μL of $H_2O$. The thermal cycling conditions of 16 S rRNA: 95 °C (3 min), 40 cycles at 95 °C (30 s), 55 °C (30 s) and 72 °C (45 s), and 72 °C (10 min). The thermal cycling conditions of ITS: 94 °C (10 min), 40 cycles at 94 °C (30 s), 56 °C (30 s) and 72 °C (45 s), and 72 °C (7 min). Samples with sterile pure water were regarded as negative control. Amplification efficiencies were 106.8–115.5% with $R^2 > 0.995$. Based on the Ribosomal RNA Operon Copy Number Database, microbial cell numbers could be assessed by the absolute abundances of 16S rRNA/4.1[93,94].

High-throughput qPCR-based SmartChip technique[95] was applied to detect denitrifying genes (narG, nirS, nirK, nosZ) abundances in the plastisphere and bulk water. The mixing system included 3.1 μL of DNA sample, 24.8 μL of Mix-enzyme (LightCycler SYBR Green I), and 3.1 μL of denitrifying gene primers. They were dispensed into a SmartChip via a MultiSample NanoDispenser, and then quantified with a Real-Time PCR system (WaferGen, Biosystems, USA). The bacterial 16 S rRNA (515 F/907 R) gene was set as a reference. The thermal cycle of qPCR conditions included heating at 95 °C (10 min), followed by 40 cycles at 95 °C (30 s), annealing at 58 °C (30 s), and extension at 72 °C (30 s). The threshold cycle of 31 was regarded as the detection limit. Information of primer pairs and other procedures are listed in our previous study[95] and Supplementary Table 4. Relative copy numbers of denitrifying gene and16S rRNA gene were estimated as follows:

$$\text{Relative gene copy numbers} = 10^{(31-\text{threshold cycle})}/(3.33) \tag{15}$$

Normalized relative denitrifying gene abundance was the proportion of the relative functional gene copy number to the 16S rRNA gene copy number.

**Statistical analysis**. In this study, the significance analyses on raw data were performed in IBM SPSS (Version 22.0). The significant differences in microbial activity, EPS concentration, c-di-GMP, and lipid/fatty acid level, $N_2O$ concentration, denitrification rate, the proportion of denitrification pathways to $N_2O$ production, functional gene abundance, denitrifiers-based niche breadth and overlap between plastisphere and bulk water were tested with one-way ANOVA method followed by a post-hoc Tukey test. $P$ value < 0.05 was considered as significant.

**Reporting summary**. Further information on research design is available in the Nature Research Reporting Summary linked to this article.

## Data availability

The sequence data generated in this study have been deposited in NCBI under accession NO. SUB10278422 (https://www.ncbi.nlm.nih.gov). The raw data of C–D ratios, lipids/fatty acids intensities, EPS, c-di-GMP, $N_2O$ production, denitrification rate, percentage, isotope ratios, microbial communities generated in this study have been provided in the

Source Data file and also been uploaded in Dryad database https://doi.org/10.5061/dryad.47d7wm3gz. Other data are available in Supporting Information. Source data are provided with this paper.

## Code availability

The codes of this study are available in Supporting Information.

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

## Acknowledgements

We thank Prof. Xinping Chen for the constructive comments on this manuscript. This project was supported by the Natural Science Foundation of China (42003060, by Xiaoxuan Su), and Science Fund for Creative Research Groups of the National Natural Science Foundation of China (42021005, by Yong-guan Zhu).

## Author contributions

X.X.S., L.Y.Y., Y.J.T., K.Y., J.L.P., H.L., and Y.-G.Z. conceived the study, designed and executed the incubations, analyzed the results, and wrote the manuscript. X.X.S. and Y.M.W. analyzed the community data. X.X.S. and L.Y.Y. measured N speciation. T.W. and H.L. detected $N_2O$ isotopes. X.X.S. conducted single-cell Raman analysis. Y.J.T., L.R., Y.-G.Z., and M.R. edited the manuscript. All authors are involved in the interpretation of the results and the preparation of this manuscript.

## Competing interests

The authors declare no competing interests.
