## [Peer Review File · Nature Communications]

Estuarine plastisphere as an overlooked source of N₂O productionEditorial Note: Parts of this Peer Review File have been redacted as indicated to maintain the confidentiality of unpublished data.

REVIEWER COMMENTS

Reviewer #1 (Remarks to the Author):

In the manuscript “Estuarine plastisphere as a potential hotspot for N₂O production”, the authors uncovered that the plastisphere was an unrecognized N₂O source and explored the related mechanism based on a combined in situ and lab-scale incubation experiment. The study is interesting and the manuscript is relatively well-organized.

Although the manuscript needs to be improved, I would love to recommend it to be accepted after a major revision.

Major concerns:

1. The authors conducted the in situ incubation experiment at a water depth of 4m, why? It is known that most plastics float on the water surface, and the densities of the four types of plastics chosen by the authors are all less than the density of water, too. Moreover, the water depth directly affects the oxygen concentration in water and affects the microbial community structure and related reactions. So, I’m concerned about whether the findings of this study could represent the conditions in the real world.
2. Biofilms on the plastics were repeatedly washed until the biofilms were detached, and then the plastisphere suspensions were incubated for the determination of N₂O concentration and isotopocules, denitrifier community, and denitrifying gene abundance. Since the growth pattern (attached or free) has significant impacts on microorganisms, why did the authors wash the biofilms off the plastics and then incubate them instead of incubating the plastics carrying the biofilms directly? Could the present results reflect the real condition?
3. It seems there are only three bulk water samples for co-occurrence network analysis. Co-occurrence network analysis is based on correlation analysis, and only three samples are obviously not suitable for correlation analysis. The authors may explain it more, or, it is suggested to add samples or delete corresponding results.

Detailed comments:

L55-56: Several studies have provided evidence for the role of the plastisphere in biogeochemical cycling, so the current expression is not rigorous.

L95-96: The third hypothesis lacks reasoning. Why did you assume that the FD and CD processes are more important than the BD process?

L112: The sentence is wrong. “and receives and” ?

L321, 328, 329: ITS, ACE and ANOSIM do not have full names when they first appeared in the paper.

L325: “denitrifiers communities” may be “denitrifier communities.”

L331: There’s an extra comma.

L577-578: The main finding of this study is that the plastisphere is a hotspot of N₂O, but here it is stated that the plastisphere has the potential to decelerate global warming, which is somewhat contradictory. Since N₂O can be produced during BD, the impact on global warming might not be “mitigation”.

L579: This study was conducted in an estuarine ecosystem, but it is the “marine” that is emphasized many times in the paper. Estuarine and marine ecosystems are different, so please do not mix them up.

Fig.S7B: The letters labeled are beyond my understanding. The “d” should be a “c”? Other figures should also be checked carefully.

Reviewer #2 (Remarks to the Author):

This manuscript reports on denitrification and n₂o production by microbial samples that derived from plastic surfaces that had been incubated in a Xiamen estuary or from the surrounding waters. All incubations were done under anoxic (He purged) conditions. The authors in addition perform a suite of additional chemical and molecular analysis on the communities.

The authors use a wide variety of complimentary techniques to study denitrification, N₂O production, and examine the microbiome.

I have strong reservation about the interpretations however – they authors claim that the ‘plastisphere exhibits higher denitrifying activity and N₂O production’ – however, the normalization of these rates is neither explained nor justified. Indeed - in the vials where they placed either ‘cells displaced from the plastic surface’ or the ‘bulk water’ – they observe consistently increased rates in vials with cells derived from the plastic surfaces. But – what is the rationale for choosing these amounts/volumes? What ‘bulk water volume’ corresponds with what ‘plastic surface area’ to make such comparisons – clearly, if a local bulk volume is taken – then there will be fewer cells (and lower activity in any process) compared to the cells present on a adjacent colonized surface. I believe this to a crucial matter to be resolved for this manuscript –but not sure it is easy to do so (in an estuary of a certain volume – how much surface area is occupied by plastic debris?) – making the claims of this manuscript ill-founded. Actual cell numbers present in either condition should be reported, and a strong justification for their choice is needed.

I also have a reservation regarding the molecular analysis of the different microbial communities – and claims regarding ‘abundances’ of different functional genes – it appears that only single primer pairs were used for the different genes. Yet, there is evidence of substantive diversity in all the genes involved

in the denitrification pathway. Therefore, seeing different abundances of different genes across different microbiomes – may also reflect slight shifts in the diversity of said genes. Hence, the authors should rigorously evaluate the coverage of the employed PCR primers given the target microbiomes (which would normally require metagenomics analysis).

The conditions under which the authors conducted the study were anoxic (purging with He) – and under those conditions, denitrification (as well as N₂O) production is substantial. Yet, they measured that the studied estuary had – at the examined depth – a DO level of ca. 1 mg/L. Would it therefore not be more appropriate to simulate those conditions? What are the authors ‘testing’ re. the totally anoxic conditions? It also appears that NO carbon source is added to the incubations? When environmental conditions become anoxic it is because ‘oxygen consuming substances’ are present in the water – they will result in bacterial oxygen consumption. But – they will also drive denitrification – and their availability – as other studies have shown – can greatly affect the extent of N₂O production. Was the carbon-source availability examined/controlled? Hence – the observations made by the authors might be strongly biased/influenced by the carbon availability in the experiments.

Finally, I find the measurements regarding the ‘biofilm’ chemical composition insufficient to make strong statements regarding the mechanisms of biofilm formation (as suggested in figure ...) - certainly the authors can detect EPS substances (yet, it is not clear how they normalize the ‘bulk water measurement), and measure c-di-GMP; yet they don’t offer insight in mechanisms. The Raman based profiling regarding ‘activity’ – is strong and indicates that the plastic surface derived cells have higher metabolic (which seems more appropriate than phenotypic) activity than bulk cells – yet, the profiling regarding ‘intracellular fatty acid/lipid levels’ are not very conclusive – yes, some differences are found with the bulk water cells – but this is merely descriptive.

Reviewer #3 (Remarks to the Author):

The manuscript submitted to Nature Communications "Estuarine plastisphere as a potential hotspot for N₂O production" reports for the first time that the plastisphere can harbor denitrification and could be a source of nitrous oxide (N₂O, a potent greenhouse gas) to the atmosphere. Some very nice isotope labeling analyses were performed and support these findings.

The implications are very interesting and important here: in many ways plastic surfaces can be envisioned as an extension of the benthos and could short circuit nitrification pathways in eutrophic estuarine environments. How important this is as a planetary process future studies are necessary to determine.

Major points:

At ~7000 words the main text is lengthy (Nature Communications limit is ~5k words) —this manuscript is on the long side here and needs to be pared down and focused- some suggestions:

-Figure 2 detailing the results of the cyclic di-GMP assays could be moved to the Supplemental section as this does not provide direct support of the main findings that members of the plastisphere are involved in denitrification.

-Figure 4 details data that are tenuous as stands. How are these quantified and normalized? Does 12 liters of bulk water microbes equal the biofilm that was harvested? How can the 16S gene that is known to vary by an order of magnitude among bacteria, be used to quantify cells? These questions are the burden to overcome here if these results are to be left as is. It is suggested that these findings be reported more as a general trend and not a quantitative fact. Scaling back these claims should not deter from the main impact of the overall findings.

Minor points:

Lines 142-143: Is this a citation of the method here? Explain. It seems odd because washing will not remove the biofilm- the really would require agitation here. Were the washed plastic surfaces assayed for adhered cells afterward? It has been shown that di-valent cations help adhere cells but only filter sterile seawater is used here—please provide some details as to how this was done and assurances were made that a bulk of the cells were removed for assay—or state the caveats involved here and the limitations of the study in this regard.

Lines 155-159: What about sediments themselves? Would they contribute to the N₂O budget? Please address this here and in the Discussion section.

Replies and Explanations to Comments

We thank the reviewers for their constructive comments. We have conducted additional experiments and expanded our data analyses as suggested by the reviewers. We have now revised the manuscript thoroughly and highlighted the changes in track-change mode (please see “Related Manuscript File”— “Revised manuscript (track change)”). All questions raised by the reviewers have been answered point-by-point below.

Reviewer #1 Comments:

In the manuscript “Estuarine plastisphere as a potential hotspot for N₂O production”, the authors uncovered that the plastisphere was an unrecognized N₂O source and explored the related mechanism based on a combined in situ and lab-scale incubation experiment. The study is interesting and the manuscript is relatively well-organized. Although the manuscript needs to be improved, I would love to recommend it to be accepted after a major revision.

Major comments:

1. The authors conducted the in situ incubation experiment at a water depth of 4m, why? It is known that most plastics float on the water surface, and the densities of the four types of plastics chosen by the authors are all less than the density of water, too. Moreover, the water depth directly affects the oxygen concentration in water and affects the microbial community structure and related reactions. So, I’m concerned about whether the findings of this study could represent the conditions in the real world.

Answer:

Thanks. We conducted the in situ incubation at the water depth of 4m for the following reasons.

Firstly, the distribution of marine plastic debris covers a wide range, from the surface to the water column and the sediment (please see figure below)[1, 2]. It has been estimated that only a small proportion (<0.3 million metric tons) of plastic debris is currently circulating on the marine surface[3]. By contrast, the remaining large proportions of plastic debris would be in suspension or quickly transferred to the water column and the deep ocean by weathering and sinking[4].

[Redacted]

Secondly, to prevent loss of plastic samples during the 30d in situ incubation, we connected these plastics with cotton cords and used stone debris placed under the plastics to maintain buoyancy. This thus led to a ~4m water depth for the plastics suspended in the estuarine water.

Given all that, the in situ incubation was conducted at the water depth of ~4m. Furthermore, all the lab-scale experiments were based on the in situ conditions at ~4m (water quality, dissolved oxygen concentration, etc., see Methods). Therefore, we believe that the results could represent the in situ estuarine conditions. To avoid

confusion, we have revised and made updates.

".....the obtained 10×10cm plastics debris was sterilized using 70% ethanol and connected with cotton cords, and we used stone debris placed under the plastics to maintain buoyancy. This led to a ~4m water depth for the plastics suspended in estuarine water." (Page 32 Lines 694-696).

2. Biofilms on the plastics were repeatedly washed until the biofilms were detached, and then the plastisphere suspensions were incubated for the determination of N₂O concentration and isotopocules, denitrifier community, and denitrifying gene abundance. Since the growth pattern (attached or free) has significant impacts on microorganisms, why did the authors wash the biofilms off the plastics and then incubate them instead of incubating the plastics carrying the biofilms directly? Could the present results reflect the real condition?

Answer:

Thanks. In this study, the biofilms on the plastics were washed and detached to form plastisphere suspensions, aiming to conduct the subsequent 24h lab-scale experiment under more controlled conditions. This is because these bulk plastics cannot be directly placed into the 120mL serum bottles used in the 24h experiment, due to the narrow opening (1.25cm) of these bottles. Besides, cutting plastics to fit into the bottles would cause a heavy loss of biofilms or biomass on the plastics. For this reason, we prepared the plastisphere suspensions in this study.

To examine if the growth pattern (attached or detached biofilms) has significant impacts on microorganisms during the 24h experiment, we have now performed additional experiments to explore N₂O production, anoxic microbial activities, and community structure of the attached and detached biofilms (please see figure

below).

Response to comments Fig. 2 (now included in the Supporting Information, **Supplementary Fig. 9**). N₂O production, anoxic microbial activities and denitrifier community compositions in attached and detached biofilms after the 24h lab-scale experiment. **(a)**. N₂O production. **(b)**. Ratios of (C-D)/(C-D+C-H) calculated by D₂O-single cell Raman technique. **(c-d)**. Phylum levels of bnirS-type and bnirK-type denitrifiers. **(e-f)**. Genus levels of bnirS-type and bnirK-type denitrifiers. This supplemented experiment was conducted in 250mL conical flasks, because the plastic debris cannot be placed in the 120mL serum bottles.

The above results indicate that during the 24h experiment, N₂O production,

anoxic microbial activities and community structure showed no significant differences between attached and detached biofilms. This suggests that denitrifier activities during 24h were not likely to be significantly impacted by growth patterns. Therefore, we believe that the plastisphere suspensions used in this study closely reflect real conditions. These results and Response to comments Fig. 2 have now been added in the revised manuscript.

"In this study, we prepared the plastisphere suspensions, aiming to explore the biofilms on the plastics under more controlled conditions during the lab-scale experiments. To examine if the growth pattern (attached or detached biofilms) has significant impacts on microorganisms during the 24h experiment, we further investigated N₂O production, anoxic microbial activities, and community structure of the attached and detached biofilms (Supplementary Fig. 9). These results showed no significant differences between attached and detached biofilms, suggesting that denitrifier activities during 24h were not likely to be significantly impacted by growth patterns." (Pages 33-34 Lines 723-731).

3. It seems there are only three bulk water samples for co-occurrence network analysis. Co-occurrence network analysis is based on correlation analysis, and only three samples are obviously not suitable for correlation analysis. The authors may explain it more, or, it is suggested to add samples or delete corresponding results.

Answer:

Thanks. We have now reconducted the 24h experiment with more bulk water samples (5 samples), and re-employed co-occurrence network analysis of denitrifiers in surrounding bulk water (please see figure below).

Response to comments Fig. 3 (now updated in the revised manuscript, **Main Text Fig. 6**). Co-occurrence network analysis illustrating the keystone denitrifiers in the plastisphere (**a**) and bulk water (**b**).

Although some keystone bacterial and fungal denitrifiers in the bulk water shifted compared with the previous version, the main results are still the same; that is, bnirS-type denitrifiers are the core residents in bulk water.

We have now updated the figures (including network, α , β diversity, composition, niche breadth, N₂O production, denitrification rate, etc.) and the related parts in the revised manuscript.

".....the bulk water group (in quintuplicate) was set as control." (Page 33 Line 715).

Detailed comments:

4. L55-56: Several studies have provided evidence for the role of the plastisphere in biogeochemical cycling, so the current expression is not rigorous.

Answer:

Thanks. Corrected.

"However, the crucial roles of the plastisphere in biogeochemical cycling in marine environments are largely unknown." (Page 3 Line 58).

5. L95-96: The third hypothesis lacks reasoning. Why did you assume that the FD and CD processes are more important than the BD process?

Answer:

Thanks. This is because (1) recent studies suggest that a high abundance of fungi could be attached to the plastic surfaces in aquatic environment^[5]; (2) fungal denitrifiers generally lack N₂O reductase, leading to N₂O as the end product during fungal denitrification (FD); (3) during chemodenitrification (CD), iron (II) reacts with the reactive nitrogen to increase N₂O production, which may also be a great contributor. In addition, previous studies conducted in sediments and soils have shown that FD and CD could contribute to significant emissions of N₂O relative to bacterial denitrification (BD)^[6-8]. Thus, we hypothesize that FD and CD may also have a large contribution in the plastisphere and water.

To avoid confusion, we have now added some statements regarding the reasoning of the third hypothesis in the Introduction section.

"Recent studies suggest that a high abundance of fungi, such as parasitic and saprophytic ones, could be attached to the plastic surfaces in aquatic environment³⁵. Moreover, fungal denitrifiers generally lack N₂O reductase, leading to N₂O as the end product^{35,36}" (Page 4 Lines 84-86).

".....N₂O production from abiotic denitrification, which may also be a great contributor relative to BD³⁷⁻³⁹" (Page 5 Line 92).

6. L112: The sentence is wrong. "and receives and" ?

Answer:

Thanks. Corrected.

".....and receives 1100mm....." (Page 31 Line 677).

7. L321, 328, 329: ITS, ACE and ANOSIM do not have full names when they first appeared in the paper.

Answer:

Thanks, we have now provided their full names.

".....Internally Transcribed Spacer (ITS) gene" (Page 42 Line 905).

".....abundance-based coverage estimation (ACE)" (Page 42 Line 913).

".....analysis of similarities (ANOSIM)....." (Page 42 Line 915).

8. L325: "denitrifiers communities" may be "denitrifier communities."

Answer:

Thanks. Corrected.

".....denitrifier communities..... " (Page 42 Line 916).

9. L331: There's an extra comma.

Answer:

Thanks. Deleted.

"Co-occurrence network analysis was performed in R^{91} (nodes $R^2 > 0.90$ and $P < 0.05$) and visualized by Gephi (Version 0.9.2)." (Page 42 Lines 916-917).

10. L577-578: The main finding of this study is that the plastisphere is a hotspot of N_2O , but here it is stated that the plastisphere has the potential to decelerate global warming, which is somewhat contradictory. Since N_2O can be produced during BD, the impact on global warming might not be "mitigation".

Answer:

Thanks. We have now deleted the sentence.

11. L579: This study was conducted in an estuarine ecosystem, but it is the “marine” that is emphasized many times in the paper. Estuarine and marine ecosystems are different, so please do not mix them up.

Answer:

Thanks. We have now corrected the word “marine” to “estuarine”.

"Fungal contributions to N₂O production in estuarine water have been less well studied." (Page 28 Line 613).

"Plastisphere derived from the widespread dissemination of plastic debris in estuarine environments....." (Page 30 Line 655).

"the importance of estuarine plastisphere in N₂O production." (Page 31 Line 663).

12. Fig.S7B: The letters labeled are beyond my understanding. The “d” should be a “c”? Other figures should also be checked carefully.

Answer:

Thanks. We have corrected and checked both the Main Text and the Supporting Information.

Reviewer #2 Comments:

This manuscript reports on denitrification and N₂O production by microbial samples that derived from plastic surfaces that had been incubated in a Xiamen estuary or from the surrounding waters. All incubations were done under anoxic conditions. The authors in addition perform a suite of additional chemical and molecular analysis on the communities. The authors use a wide variety of complimentary techniques to study denitrification, N₂O production, and examine the microbiome.

13. I have strong reservation about the interpretations however – the authors claim that the ‘plastisphere exhibits higher denitrifying activity and N₂O production’ – however, the normalization of these rates is neither explained nor justified. Indeed - in the vials where they placed either ‘cells displaced from the plastic surface’ or the ‘bulk water’ – they observe consistently increased rates in vials with cells derived from the plastic surfaces. But – what is the rationale for choosing these amounts/volumes? What ‘bulk water volume’ corresponds with what ‘plastic surface area’ to make such comparisons – clearly, if a local bulk volume is taken – then there will be fewer cells (and lower activity in any process) compared to the cells present on an adjacent colonized surface. I believe this to a crucial matter to be resolved for this manuscript –but not sure it is easy to do so (in an estuary of a certain volume – how much surface area is occupied by plastic debris?) – making the claims of this manuscript ill-founded. Actual cell numbers present in either condition should be reported, and a strong justification for their choice is needed.

Answer:

Thanks for this excellent comment, which we address in detail below.

(1) **Point 1: Normalization of N₂O production and denitrification rate.** In this study, the plastisphere biofilms were repeatedly washed with a certain volume (40mL) of sterile estuarine water until a large proportion of the biofilms were detached from plastics. This formed plastisphere suspensions (in liquid phase), which were then used in the following lab-scale experiments. Thus, the results of plastisphere were expressed on a per volume water basis. We have supplemented experiments to confirm that microbial growth patterns (i.e., attached or detached biofilms) had no significant impacts on denitrifier activities during 24h. The detailed results are shown

in Response to Comments 2. In addition, the reason for the selection of 40mL water volumes is provided below in **Point 2**.

In consideration of the reviewer's comments, we have now provided the absolute abundances of 16S rRNA and ITS genes of the plastisphere and surrounding bulk water (please see figure below). We find that both the abundances of bacteria and fungi are significantly higher in the plastisphere than those in bulk water ($P < 0.001 \sim 0.027$).

Response to comments Fig. 4 (now included in the Supporting Information, **Supplementary Fig. 5**). The absolute abundances of 16S rRNA and ITS genes in the plastisphere and bulk water.

Different letters (a, b, c, and d) indicate the significant differences ($P < 0.05$) among each plastisphere group and bulk water group.

During revisions, we have normalized the denitrification rate and N₂O production with microbial biomasses (referring to cell numbers) in the plastisphere and bulk water (please see figure below). Based on the Ribosomal RNA Operon Copy Number Database, the average microbial cell numbers could be estimated by the absolute abundances of 16S rRNA/4.1 [9, 10]. The results after normalization reinforce the fact that the plastisphere exhibits higher denitrification rates and N₂O productions compared with surrounding bulk water ($P < 0.001 \sim 0.042$). This is consistent with our results of the previous version, although the N₂O production of

PS was less than the other plastisphere ($P=0.012$).

Response to comments Fig. 5 (now updated in the revised manuscript, **Main Text Fig. 4**).

Denitrifying activity and N₂O production in the plastisphere and bulk water. **a.** Concentrations of N₂O after the 24h denitrification experiment. **b.** Denitrification rates quantified by ¹⁵N isotope pairing technique. Both N₂O production and denitrification rate are normalized with microbial cell numbers by using the absolute abundances of 16S rRNA in the plastisphere and bulk water (see Methods). Different letters (a, b, and c) indicate the significant differences ($P < 0.05$) among each plastisphere group and bulk water group.

The two figures have been added in the revised manuscript (Main Text or Supporting Information), and the related parts have been updated.

"After 24h, N₂O concentrations were 0.44~0.59 fmol cell⁻¹ in the plastisphere, greater than 0.41 fmol cell⁻¹ in bulk water ($P < 0.001 \sim 0.042$, Fig. 4a). The N₂O concentrations in the PS plastisphere were lower than the other types of plastisphere ($P=0.012$). For denitrification rate, the plastisphere registered higher rates ranging from 3.36 to 4.07 fmol cell⁻¹ h⁻¹, while the bulk water had a lower rate (mean value 2.97 fmol cell⁻¹ h⁻¹) ($P=0.001$, Fig. 4b)."(Page 20 Lines 425-432).

"We first quantified the abundances of 16S rRNA and ITS genes in the plastisphere and bulk water (Supplementary Fig. 5), and found that the abundances of bacteria and fungi were higher in the plastisphere than those in bulk water ($P < 0.001 \sim 0.027$)."(Page 22 Lines 461-463).

"The abundances of bacteria (16S rRNA) and fungi (ITS) in the plastsphere and bulk water were quantified with qPCR technique using the universal primers 515F/907R and ITS1F/ITS2R, respectively^{68,91}. The mixing system (20 μ L) included 2 μ L of DNA sample, 10 μ L of Mix-enzyme (LightCycler SYBR Green I), 0.8 μ L of 515F/907R or ITS1F/ITS2R primers, and 7.2 μ L of H₂O. The standard curves were obtained by a serial dilution of the plasmids with 16S rRNA and ITS genes. The thermal cycling conditions of 16S rRNA: 95°C (3min), 40 cycles at 95°C (30s), 55°C (30s) and 72°C (45s), and 72°C (10min). The thermal cycling conditions of ITS: 94°C (10min), 40 cycles at 94°C (30s), 56°C (30s) and 72°C (45s), and 72°C (7min). Samples with sterile pure water were regarded as the negative control. Amplification efficiencies were 106.8%–115.5% with $R^2 > 0.995$. Based on the Ribosomal RNA Operon Copy Number Database, microbial cell numbers could be estimated by the absolute abundances of 16S rRNA/4.1."(Page 43 Lines 928-939).

"Both N₂O production and denitrification rate are normalized with microbial cell numbers by using the absolute abundances of 16S rRNA in the plastsphere and bulk water (see Methods)." (Page 56 Lines 1245-1248).

(2) **Point 2: Reason for the selection of 40mL water volumes.** While preparing the experiment, we found that the minimum water volume for detachment of biofilms from plastic debris (10cm \times 10cm) was 40mL. Thus, we selected this water volume to conduct the following experiment in this study. To avoid bias induced by the selected water volume used for incubations, we have conducted additional experiments with different water volumes (80 mL and 120 mL) (please see figure below). We found a similar result that N₂O production after 24h in the plastsphere was higher than that in bulk water, regardless of water volumes. This indicates that the plastsphere could indeed act as a hotspot for denitrification and N₂O production, compared to surrounding bulk water.

Based on our data, we infer the plastisphere as a hotspot for denitrification and N₂O production mainly for two reasons: (1) the capacity of N₂O production per cell in the plastisphere is higher than that in bulk water, probably owing to the growth patterns (attached or free-living) and microenvironments (anoxic or oxic-anoxic); (2) the biomass attached to the surface of plastics is greater than in surrounding bulk water, thereby leading to the hotspot. To avoid confusion, we have now clarified the reasoning behind our water volume selection in the revised manuscript.

Response to comments Fig. 6 (now included in the Supporting Information, **Supplementary Fig. 10**). N₂O productions in the plastisphere and bulk water with different water volumes. **a.** 80 mL. **b.** 120 mL. Both are normalized with microbial biomasses in the plastisphere and bulk water.

Different letters (a, b, and c) indicate the significant differences ($P < 0.05$) among each plastisphere group and bulk water group.

"In addition, we selected the water volume of 40mL to conduct the 24h experiment, because it is the minimum volume for a large proportion of biofilm detachment from plastics debris. We also conducted the same experiments with different volumes (80 mL and 120 mL) to avoid bias induced by incubation water volume (Supplementary Fig. 10). The results showed that N₂O production in the plastisphere was higher than that in bulk water, regardless of water volumes."(Page

34 Lines 731-736).

14. I also have a reservation regarding the molecular analysis of the different microbial communities – and claims regarding ‘abundances’ of different functional genes – it appears that only single primer pairs were used for the different genes. Yet, there is evidence of substantive diversity in all the genes involved in the denitrification pathway. Therefore, seeing different abundances of different genes across different microbiomes – may also reflect slight shifts in the diversity of said genes. Hence, the authors should rigorously evaluate the coverage of the employed PCR primers given the target microbiomes (which would normally require metagenomics analysis).

Answer:

Thanks. During our preliminary experiments, the common and widely reported primers of genes nirS and nirK were selected to conduct PCR assays[11-14]. Information on the primers are listed below:

nirS

- **Cd3aF** G TSAACG TSAAGGARACSSG / **R3cd** GASTTCGGRTGSGTCTTGA
- **nirSC1F** ATCGTCAACGTCAARGARACVGG/**nirSC1R** TTCGGGTGCGTCTTSABGAASAG
- **nirSC2F** TGGAGAACGCCGGNCARGTNTGG/ **nirSC2R** GATGATGTCCACGGCNACRT

nirK

- **F1aCu** ATCATGGTCTGCCGCG / **R3Cu** GCCTCGATCAGRTRTGGTT
- **876C** ATYGGCGGVCA YGGCGA / **1040** GCCTCGATCAGRTRTGGTT
- **nirK1F** GGMATGGTKCCSTGGCA / **nirK5R** GCCTCGATCAGRTRTGGTT

In PCR assays, the target bands of PCR products were successfully amplified only for Cd3aF/R3cd (nirS) and F1aCu/R3Cu (nirK) primers. The other primers were not amplified successfully (please see figures below). Thus, we chose to apply the primers

Cd3aF/R3cd and F1aCu/R3Cu in this study to explore denitrifier communities.

Furthermore, these two primer pairs (Cd3aF/R3cd and F1aCu/R3Cu) have been regarded as the universal primers for targeting denitrifiers in environments, which have been widely used in water, soil and sediment ecosystems[11, 13-16]. The target efficiencies and the coverages of these two primers are high (over 70%)[17, 18], and thus they are representative for exploring denitrifier communities in various ecosystems.

Response to comments Fig. 7. The agarose gel electrophoresis of DNA with Cd3aF/R3cd primers (426 bp) targeting nirS gene. Of these, 1-3 represent the PE plastisphere, 4-6 represent the PS plastisphere, 7-9 represent the PP plastisphere, 10-12 represent the PVC plastisphere, and 13-15 represent the surrounding bulk water. CK is a negative control with only sterile pure water. The mixing system (25 μ L) included 2 μ L of DNA sample, 0.5 μ L of BSA, 12.5 μ L of Mix-enzyme (2X Pro Taq Master Mix), 1.2 μ L of primers, and 8.8 μ L of H₂O. The thermal cycle of qPCR conditions included heating at 95°C (3min), followed by 32 cycles at 95°C (30s), annealing at 57°C (30s), and extension at 72°C (45s).

Response to comments Fig. 8. The agarose gel electrophoresis of DNA with F1aCu/R3Cu primers (450 bp) targeting nirK gene. Of these, 1-3 represent the PE plastisphere, 4-6 represent the PS plastisphere, 7-9 represent the PP plastisphere, 10-12 represent the PVC plastisphere, and 13-15 represent the surrounding bulk water. CK is a negative control with only sterile pure water.

Replies and Explanations to Comments

13-15 represent the surrounding bulk water. CK is a negative control with only sterile pure water. The mixing system (25 μ L) included 2 μ L of DNA sample, 0.5 μ L of BSA, 12.5 μ L of Mix-enzyme (2X Pro Taq Master Mix), 1.2 μ L of primers, and 8.8 μ L of H₂O. The thermal cycle of qPCR conditions included heating at 95°C (3min), followed by 32 cycles at 95°C (30s), annealing at 60°C (30s), and extension at 72°C (45s).

Response to comments Fig. 9. The agarose gel electrophoresis of DNA with nirSC1F/nirSC1R primers targeting nirS gene. Of these, 1-3 represent the PE plastisphere, 4-6 represent the PS plastisphere, 7-9 represent the PP plastisphere, 10-12 represent the PVC plastisphere, and 13-15 represent the surrounding bulk water. CK is a negative control with only sterile pure water. The thermal cycle of qPCR conditions included heating at 95°C (10min), followed by 32 cycles at 95°C (30s), annealing at 57°C (30s), and extension at 72°C (45s).

Response to comments Fig. 10. The agarose gel electrophoresis of DNA with nirSC2F/nirSC2R primers targeting nirS gene. Of these, 1-3 represent the PE plastisphere, 4-6 represent the PS plastisphere, 7-9 represent the PP plastisphere, 10-12 represent the PVC plastisphere, and 13-15 represent the surrounding bulk water. CK is a negative control with only sterile pure water. The thermal cycle of qPCR conditions included heating at 95°C (10min), followed by 32 cycles at 95°C (30s), annealing at 57°C (30s), and extension at 72°C (30s).

Response to comments Fig. 11. The agarose gel electrophoresis of DNA with 876C/1040 primers targeting nirK gene. Of these, 1-3 represent the PE plastisphere, 4-6 represent the PS plastisphere, 7-9 represent the PP plastisphere, 10-12 represent the PVC plastisphere, and 13-15 represent the surrounding bulk water. CK is a negative control with only sterile pure water. The thermal cycle of qPCR conditions included heating at 95°C (15min), followed by 40 cycles at 98°C (10s), annealing at 60°C (30s), and extension at 72°C (30s).

Response to comments Fig. 12. The agarose gel electrophoresis of DNA with nirK1F/nirK5R primers targeting nirK gene. Of these, 1-3 represent the PE plastisphere, 4-6 represent the PS plastisphere, 7-9 represent the PP plastisphere, 10-12 represent the PVC plastisphere, and 13-15 represent the surrounding bulk water. CK is a negative control with only sterile pure water. The thermal cycle of qPCR conditions included heating at 95°C (10min), followed by 35 cycles at 95°C (30s), annealing at 60°C (30s), and extension at 72°C (30s).

15. (1) The conditions under which the authors conducted the study were anoxic (purging with He) – and under those conditions, denitrification production is substantial. Yet, they measured that the studied estuary had – at the examined depth – a DO level of <1 mg/L. Would it therefore not be more appropriate to simulate those conditions? What are the authors ‘testing’ re. the totally anoxic conditions? **(2)** It also appears that no carbon source is added to the incubations? When environmental conditions become anoxic it is because ‘oxygen consuming substances’ are present in the water – they will result in bacterial oxygen consumption. But – they will also drive denitrification – and their availability – as other studies have shown – can greatly affect the extent of N₂O production. Was the carbon-source availability examined/controlled? Hence – the observations made by the authors might be strongly biased/influenced by the carbon availability in the experiments.

Answer:

Thanks for this comment. (1) It is generally accepted that <2mg/L of dissolved oxygen concentration in water is considered as anoxic[19]. Under such conditions, denitrification-derived N₂O commonly dominates over nitrification-derived N₂O[20-22]. Therefore, this study focuses on the denitrification processes, including bacterial, fungal and chemo-denitrification. To simplify the experimental operation, we applied pure He gas to achieve the anoxic conditions. We have now included the information in the revised manuscript to clarify this point.

"During the in situ incubation in the estuary, the plastic debris was suspended in the water depth of ~4m, where the dissolved oxygen concentration was 1.19mg L⁻¹ (<2.0mg L⁻¹) and is generally considered as anoxic⁴⁴. Under such conditions, denitrification-derived N₂O commonly dominates over nitrification-derived N₂O^{45,46}.

Therefore, we focused on denitrification processes in this study." (Page 20 Lines 419-422).

(2) Yes, no additional carbon source was added during the lab-scale incubation.

This is because we aimed to investigate the in situ or basal capacities of denitrification and N₂O production of the plastsphere and the bulk water. In this study, the water for lab-scale incubation (plastsphere and bulk water groups) was taken from in situ estuarine water, in which the organic matter concentrations are 21.2~25.7mg L⁻¹. This means that the estuarine water contains sufficient carbon

sources for denitrification. Therefore, our results likely did reflect the actual denitrification capacity in the estuarine conditions, and was unlikely biased by the carbon availability. We have now updated the related statements to avoid confusion.

"No additional carbon or nitrogen source was added, aiming to simulate in situ conditions and explore the basal capacities of denitrification in the plastsphere and bulk water." (Page 33 Lines 718-720).

"The estuarine water contained 21.2~25.7mg L⁻¹ total organic carbon,....." (Page 30 Line 681).

16. Finally, (1) I find the measurements regarding the ‘biofilm’ chemical composition insufficient to make strong statements regarding the mechanisms of biofilm formation (as suggested in figure ...) - certainly the authors can detect EPS substances (yet, it is not clear how they normalize the bulk water measurement), and measure c-di-GMP; yet they don’t offer insight in mechanisms. (2) The Raman based profiling regarding ‘activity’– is strong and indicates that the plastic surface derived cells have higher metabolic (which seems more appropriate than phenotypic) activity than bulk cells – yet, the profiling regarding ‘intracellular fatty acid/lipid levels’ are not very conclusive – yes, some differences are found with the bulk water cells – but this is merely descriptive.

Answer:

Thanks for this comment.

(1) As for the mechanisms of biofilm formation, we have deleted the word "mechanism", and also corrected the units of EPS concentrations and normalized them to $\mu\text{g}/\mu\text{L}$ or $\mu\text{g}/\text{mL}$ (please see Main Text Fig. 2).

".....intracellular lipid/fatty acid levels to explore the ~~mechanisms~~ of plastisphere formation." (Page 5 Line 107).

(2) As for the descriptions of the fatty acid/lipid results, we have revised and added some discussions in the manuscript.

"Intracellular lipid/fatty acid signals are reported to be involved in biofilm formation and dispersion²⁵ Accumulations of lipid/fatty acids in microorganisms could activate the activity of phosphodiesterases (catalyzing c-di-GMP consumption pathway), leading to lower c-di-GMP levels and biofilm dispersion. Moreover, lipid/fatty acid signaling molecules such as cis-2-decenoic acid, cis-11-methyl-2-

dodecenoic acid, and cis-2-dodecenoic acid, are capable of dismantling small microaggregates and inhibiting biofilm formation^{53,54}. For example, previous studies have shown that biofilm roughness, thickness, and biomass were significantly reduced in the presence of cis-2-decenoic acid⁵³. This is likely because these signaling molecules could maintain bacterial and fungal cells metabolically active, benefit for dispersive growth. Thus, the lower intracellular lipid/fatty acid levels observed here might be another important reason for the biofilm formation on the plastics." (Page 25 Lines 528-543).

Reviewer #3 Comments:

The manuscript submitted to Nature Communications "Estuarine plastisphere as a potential hotspot for N₂O production" reports for the first time that the plastisphere can harbor denitrification and could be a source of nitrous oxide (N₂O, a potent greenhouse gas) to the atmosphere. Some very nice isotope labeling analyses were performed and support these findings.

The implications are very interesting and important here: in many ways plastic surfaces can be envisioned as an extension of the benthos and could short circuit nitrification pathways in eutrophic estuarine environments. How important this is as a planetary process future studies are necessary to determine.

Major points:

17. At ~7000 words the main text is lengthy (Nature Communications limit is ~5k words) —this manuscript is on the long side here and needs to be pared down and focused- some suggestions:

-Figure 2 detailing the results of the cyclic di-GMP assays could be moved to the Supplemental section as this does not provide direct support of the main findings that members of the plastisphere are involved in denitrification.

Answer:

Thanks for the suggestion. We have modified Figure 2 and moved the part of cyclic-di-GMP to Supplementary Information.

Response to comments Fig. 13 (now updated in the revised manuscript, **Main Text Fig. 2**).

Concentrations of extracellular polymeric substances (EPS) in the plastisphere and bulk water. **a.** Schematic overview of biofilm formation and dispersion on plastic surfaces in water, including cell initial attachment, irreversible attachment, mature and dispersion. This diagram is modified from previous studies^{25,49,95}. **b-d.** Concentrations of extracellular polysaccharide (PS), extracellular protein (PN) and extracellular DNA (eDNA). Different letters (a, b, and c) indicate the significant differences ($P < 0.05$) among each plastisphere group and bulk water group.

Response to comments Fig. 14 (now included in the Supporting Information, **Supplementary Fig. 3**). Concentrations of intracellular cyclic di-GMP (c-di-GMP) in the plastisphere and bulk water. N.D. is non-detectable. Different letters (a and b) indicate the significant differences ($P < 0.05$) among each plastisphere group and bulk water group.

18. -Figure 4 details data that are tenuous as stands. How are these quantified and normalized? Do 12 liters of bulk water microbes equal the biofilm that was harvested? How can the 16S gene that is known to vary by an order of magnitude among bacteria, be used to quantify cells? These questions are the burden to overcome here if these results are to be left as is. It is suggested that these findings be reported more as a general trend and not a quantitative fact. Scaling back these claims should not deter from the main impact of the overall findings.

Answer:

Thanks for this comments. Please see Responses to Comment 13.

Minor points:

19. Lines 142-143: Is this a citation of the method here? Explain. It seems odd because washing will not remove the biofilm- the really would require agitation here. Were the washed plastic surfaces assayed for adhered cells afterward? It has been shown that di-valent cations help adhere cells but only filter sterile seawater is used here—please provide some details as to how this was done and assurances were made that a bulk of the cells were removed for assay—or state the caveats involved here and the limitations of the study in this regard.

Answer:

Thanks. This is in reference to the experimental design conducted in this study.

The word “wash” used here refers to “shake”, “agitation”, and then “wash”. In detail, the collected plastic debris was transferred to 250mL sterile beakers, and the biofilms or adhesive cells on the plastics were detached by shaking, agitating and washing with the sterile bulk water. We ensure that a large proportion of biofilms or adhesive cells are detached. To clarify, we have now modified the sentence.

"..... the collected plastic debris was placed in these beakers, and biofilms were repeatedly shaken, agitated, and washed with the sterile bulk water until a large proportion of the biofilms were detached." (Page 32 Lines 707-709).

20. Lines 155-159: What about sediments themselves? Would they contribute to the N₂O budget? Please address this here and in the Discussion section.

Answer:

Thanks. Yes, sediments could also contribute to N₂O emissions. As the estuarine sediments are at ~11m below water, the contribution to the N₂O budgets of the plastisphere and bulk water would be minor in this study. According to the reviewer’s suggestion, we have now added some statements to clarify in the revised manuscript.

"Although the estuarine sediments could also contribute to N₂O emissions, its effects on the plastisphere and surrounding bulk water at surface water layers would be minor. Thus, we did not consider the sediment-derived N₂O in this study. "(Page 30 Lines 649-652).

"In addition, the effects of the estuarine sediments on N₂O budgets of the plastisphere and bulk water are not considered in this study."(Page 34 Lines 741-742).

References:

1. MacLeo, M.; Arp, H. P. H.; Tekman, M. B.; Jahnke, A. The global threat from plastic pollution. *Science* 2021; 373 (6550):61-65.
2. Stubbins, A.; Law, K. L.; Munoz, S. E.; Bianchi, T. S.; Zhu, L. Plastics in the Earth system. *Science* 2021; 373 (6550):51-55.
3. van Sebille, E.; Aliani, S.; Law, K. L.; Maximenko, N.; Alsina, J. M.; Bagaev, A.; Bergmann, M.; Chapron, B.; Chubarenko, I.; Cozar, A.; Delandmeter, P.; Egger, M.; Fox-Kemper, B.; Garaba, S. P.; Goddijn-Murphy, L.; Hardesty, B. D.; Hoffman, M. J.; Isobe, A.; Jongedijk, C. E.; Kaandorp, M. L. A.; Khatmullina, L.; Koelmans, A. A.; Kukulka, T.; Laufkotter, C.; Lebreton, L.; Lobelle, D.; Maes, C.; Martinez-Vicente, V.; Maqueda, M. A. M.; Poulain-Zarcos, M.; Rodriguez, E.; Ryan, P. G.; Shanks, A. L.; Shim, W. J.; Suaria, G.; Thiel, M.; van den Bremer, T. S.; Wichmann, D. The physical oceanography of the transport of floating marine debris. *Environmental Research Letters* 2020; 15 (2).
4. Koelmans, A. A.; Kooi, M.; Law, K. L.; van Sebille, E. All is not lost: deriving a top-down mass budget of plastic at sea. *Environmental Research Letters* 2017; 12 (11).
5. Wang, L.; Tong, J.; Li, Y.; Zhu, J.; Zhang, W.; Niu, L.; Zhang, H. Bacterial and fungal assemblages and functions associated with biofilms differ between diverse types of plastic debris in a freshwater system. *Environ. Res.* 2021; 196.
6. Wankel, S. D.; Ziebis, W.; Buchwald, C.; Charoenpong, C.; de Beer, D.; Dentinger, J.; Xu, Z.; Zengler, K. Evidence for fungal and chemodenitrification based N₂O flux from nitrogen impacted coastal sediments. *Nat. Commun.* 2017; 8.
7. Zou, Y.; Hirono, Y.; Yanai, Y.; Hattori, S.; Toyoda, S.; Yoshida, N. Isotopomer analysis of nitrous oxide accumulated in soil cultivated with tea (*Camellia sinensis*) in

- Shizuoka, central Japan. *Soil Biol. Biochem.* 2014; 77:276-291.
8. Wang, M.; Hu, R. G.; Ruser, R.; Schmidt, C.; Kappler, A. Role of chemodenitrification for N₂O emissions from nitrate reduction in rice paddy soils. *Acs Earth Space Chem.* 2020; 4 (1):122-132.
 9. Zhu, D.; An, X. L.; Chen, Q. L.; Yang, X. R.; Christie, P.; Ke, X.; Wu, L. H.; Zhu, Y. G. Antibiotics disturb the microbiome and decrease the incidence of resistance genes in the gut of a common soil collembolan. *Environ. Sci. Technol.* 2018; 52 (5):3081-3090.
 10. Klappenbach, J. A.; Saxman, P. R.; Cole, J. R.; Schmidt, T. M. rrndb: the Ribosomal RNA Operon Copy Number Database. *Nucleic Acids Res.* 2001; 29 (1):181-184.
 11. Seeley, M. E.; Song, B.; Passie, R.; Hale, R. C. Microplastics affect sedimentary microbial communities and nitrogen cycling. *Nat. Commun.* 2020; 11 (1).
 12. Zheng, B.; Zhu, Y.; Sardans, J.; Penuelas, J.; Su, J. QMEC: a tool for high-throughput quantitative assessment of microbial functional potential in C, N, P, and S biogeochemical cycling. *Sci. China Life Sci.* 2018; 61 (12):1451-1462.
 13. Ye, C.; Cheng, X. L.; Zhang, K. R.; Du, M.; Zhang, Q. F. Hydrologic pulsing affects denitrification rates and denitrifier communities in a revegetated riparian ecotone. *Soil Biol. Biochem.* 2017; 115:137-147.
 14. Liu, S. Q.; Wang, C. H.; Hou, J.; Wang, P. F.; Miao, L. Z.; Fan, X. L.; You, G. X.; Xu, Y. Effects of Ag and Ag₂S nanoparticles on denitrification in sediments. *Water Res.* 2018; 137:28-36.
 15. Frostegård, Å.; Vick, S. H. W.; Lim, N. Y. N.; Bakken, L. R.; Shapleigh, J. P. Linking meta-omics to the kinetics of denitrification intermediates reveals pH-dependent causes of N₂O emissions and nitrite accumulation in soil. *The ISME Journal* 2021.
 16. de Vries, F. T.; Griffiths, R. I.; Bailey, M.; Craig, H.; Girlanda, M.; Gweon, H. S.; Hallin,

- S.; Kaisermann, A.; Keith, A. M.; Kretzschmar, M.; Lemanceau, P.; Lumini, E.; Mason, K. E.; Oliver, A.; Ostle, N.; Prosser, J. I.; Thion, C.; Thomson, B.; Bardgett, R. D. Soil bacterial networks are less stable under drought than fungal networks. *Nature Communications* 2018; 9 (1):3033.
17. Hallin, S.; Lindgren, P. E. PCR detection of genes encoding nitrite reductase in denitrifying bacteria. *Appl. Environ. Microbiol.* 1999; 65 (4):1652-1657.
18. Throbäck, I. N.; Enwall, K.; Jarvis, Å.; Hallin, S. Reassessing PCR primers targeting nirS, nirK and nosZ genes for community surveys of denitrifying bacteria with DGGE. *FEMS Microbiology Ecology* 2004; 49 (3):401-417.
19. Rabalais, N. N.; Turner, R. E.; Justić, D.; Dortch, Q.; Wiseman, W. J. Characterization of Hypoxia: Topic 1 Report for the Integrated Assessment on Hypoxia in the Gulf of Mexico. 1999; (15).
20. Wu, D.; Wei, Z.; Well, R.; Shan, J.; Yan, X.; Bol, R.; Senbayram, M. Straw amendment with nitrate-N decreased N₂O/(N₂O+N₂) ratio but increased soil N₂O emission: A case study of direct soil-born N₂ measurements. *Soil Biol. Biochem.* 2018; 127:301-304.
21. Song, X. T.; Ju, X. T.; Topp, C. F. E.; Rees, R. M. Oxygen Regulates Nitrous Oxide Production Directly in Agricultural Soils. *Environ. Sci. Technol.* 2019; 53 (21):12539-12547.
22. Yang, L.; Zhang, X.; Ju, X.; Wu, D. Oxygen-depletion by rapid ammonia oxidation regulates kinetics of N₂O, NO and N₂ production in an ammonium fertilised agricultural soil. *Soil Biology & Biochemistry* 2021; 163.

REVIEWER COMMENTS

Reviewer #1 (Remarks to the Author):

The study is interesting and the revised version has been improved a lot. All my comments have been addressed.

Reviewer #2 (Remarks to the Author):

I appreciate that the authors have updated their data presentation and now express rates normalized to the cell concentration. The authors show that the plastisphere shows a slightly higher cell-normalized N₂O production and denitrification activity than the bulk surrounding cells – and I agree with their inference that this is likely because the ‘surrounding cells’ have, on average, a lower metabolic activity. However, this small cell-normalized difference barely makes the plastisphere a ‘hotspot’ of N₂O production (as claimed throughout the manuscript and title). It is only when a large volume of the estuary volume would be filled with plastisphere – that a hotspot would be found. Hence, I still disagree with the inferences claimed by this manuscript – and I consider the title of the manuscript misleading. I believe the authors have demonstrated that plastisphere bacteria – on average- have a higher metabolic activity than bulk phase bacteria – but that the hotspot inference is unsubstantiated.

I also remain in disagreement with the author’s statement ‘it is generally assumed that less than 2 mg/L DO in water’ is considered anoxic’ (- that condition is at best ‘hypoxic’) and that ‘under such conditions ‘denitrification derived N₂O’ dominates over nitrification derived N₂O’. The citations used to support this claim refer to soil systems – where such may be the case - but I disagree that this is true in water saturated systems. Given the bold claim made in this manuscript – that under ‘in situ conditions’ the plastisphere is a hot spot - more careful control of experimental conditions to reflect in situ conditions appears warranted. I would encourage the author to conduct their experiments with in situ relevant DO concentrations to satisfy this concern. Also depending on the available carbon and ammonium – nitrifier associated N₂O production is possible. Yet – more importantly is the fact that the presence of oxygen will also affect the denitrification process itself; it has been well demonstrated in other studies that the presence of oxygen often has a substantive impact on denitrification- associated N₂O production. Citing ‘old’ papers that claim that 2 mg/L is anoxic and that under such conditions only denitrification matters is, in my opinion, not defensible. I contend that the lab-conditions did not adequately reflect the in situ conditions.

The author’s claim that sufficient organic carbon was present is debatable – as the experiments are conducted under batch conditions (and the experiment follows the depletion of the originally present nitrate – which would already consume some of the original organic carbon that is available) – while in

situ substantive fluxes of carbon would be replenish consumed carbon. Was there sufficient carbon for denitrification? The authors state the initial (in situ) concentrations; I suggest that authors also add a measurement of the carbon at the addition of the nitrate spike and the remaining organic carbon at the end of the batch experiments. This condition, again, matters as carbon availability can have a substantive effect on N₂O production.

Reviewer #3 (Remarks to the Author):

All of my previous comments and concerns have been sufficiently addressed by the authors.

Replies and Explanations to Comments

We thank the three reviewers for the constructive comments. We have now conducted additional experiments and expanded our data analyses as suggested by reviewer 2. We have revised the manuscript and highlighted the changes in track-change mode (please see “Related Manuscript File”). All questions have been answered point-by-point below.

Reviewer #1 Comments:

1. The study is interesting and the revised version has been improved a lot. All my comments have been addressed.

Answer:

We thank the reviewer for the positive feedback, and the study has been substantially improved after incorporating the reviewer’s comments.

Reviewer #2 Comments:

I appreciate that the authors have updated their data presentation and now express rates normalized to the cell concentration. The authors show that the plastisphere shows a slightly higher cell-normalized N₂O production and denitrification activity than the bulk surrounding cells – and I agree with their inference that this is likely because the ‘surrounding cells’ have, on average, a lower metabolic activity.

2. However, this small cell-normalized difference barely makes the plastisphere a 'hotspot' of N₂O production (as claimed throughout the manuscript and title). It is only when a large volume of the estuary volume would be filled with plastisphere – that a hotspot would be found. Hence, I still disagree with the inferences claimed by this manuscript – and I consider the title of the manuscript misleading. I believe the authors have demonstrated that plastisphere bacteria – on average- have a higher metabolic activity than bulk phase bacteria – but that the hotspot inference is unsubstantiated.

Answer:

Thanks, we appreciate this comment. We have modified the description of “hotspot” to that of “an overlooked source” to tone down the claim of this phenomenon throughout the revised manuscript. As the plastisphere shows a higher N₂O production and denitrifying activity than the surrounding bulk water, we think that the estuarine plastisphere is indeed an overlooked source of N₂O production.

"Title: Estuarine plastisphere as an overlooked source of N₂O production" (Page 1 Line 1).

".....suggesting an overlooked N₂O source " (Page 3 Line 36).

"These corroborated that the plastisphere could be a potential source of denitrification and N₂O production" (Page 9 Line 167).

".....functioning as a neglected source of greenhouse gas N₂O production" (Page 12 Line 234).

".....and thereby resulted in the higher denitrifying activity in the plastisphere, supporting our first hypothesis" (Page 15 Lines 285-286).

3. I also remain in disagreement with the author's statement 'it is generally assumed that less than 2 mg/L DO in water' is considered anoxic' (- that condition is at best 'hypoxic') and that 'under such conditions 'denitrification derived N₂O' dominates over nitrification derived N₂O'. The citations used to support this claim refer to soil systems – where such may be the case - but I disagree that this is true in water saturated systems. Given the bold claim made in this manuscript – that under 'in situ conditions' the plastisphere is a hot spot - more careful control of experimental conditions to reflect in situ conditions appears warranted. I would encourage the author to conduct their experiments with in situ relevant DO concentrations to satisfy this concern. Also depending on the available carbon and ammonium – nitrifier associated N₂O production is possible. Yet – more importantly is the fact that the presence of oxygen will also affect the denitrification process itself; it has been well demonstrated in other studies that the presence of oxygen often has a substantive impact on denitrification- associated N₂O production. Citing 'old' papers that claim that 2 mg/L is anoxic and that under such conditions only denitrification matters is, in my opinion, not defensible. I contend that the lab-conditions did not adequately reflect the in situ conditions.

Answer:

Thanks for this good comment. In consideration of the reviewer's comment, we have now conducted an additional experiment to explore N₂O production under anoxic (dissolved oxygen (DO) <0.5 mg L⁻¹) and hypoxic conditions (DO ~1.2 mg L⁻¹, similar to in situ DO concentration) (please see figure below). We established two groups, i.e., plastisphere and bulk water groups, with and without the addition of a nitrifier inhibitor allylthiourea. Nitrification process, including hydroxylamine

oxidation and nitrifier denitrification pathways, may contribute to N₂O production under hypoxic conditions.

Response to comments Fig. 1 (now included in the Supporting Information, **Supplementary Fig. 9**). N₂O production in the plastisphere and bulk water groups under anoxic (a) and hypoxic (b) conditions with and without adding allylthiourea. Allylthiourea is a nitrifier inhibitor and can inhibit the activity of ammonia mono-oxygenase (catalyzing ammonia to hydroxylamine) of ammonia-oxidizing archaea (AOA) and bacteria (AOB). The concentration of allylthiourea used in this study was 80µM[1]. Anoxic conditions were achieved by purging with He gas (>99.99%); Hypoxic conditions were achieved by purging with He (95%) and O₂ (5%) gases. PE plastisphere was selected as a representation of plastisphere group. Different letters (a and b) indicate the significant differences (P<0.05) between the plastisphere and bulk water groups, and the groups with and without adding allylthiourea.

We found that there was no significant difference between the groups with and without allylthiourea, indicating that N₂O production via hydroxylamine oxidation in the plastisphere and bulk water was minimal, regardless of anoxic and hypoxic (in situ) conditions. This is probably because hydroxylamine oxidation producing N₂O mainly functions under aerobic conditions[2, 3].

In contrast, nitrifier denitrification contributed to 84~206 $\mu\text{g N}_2\text{O L}^{-1}$ (Response to comments Fig. 1b), likely acting as a potential contributor in the plastsphere and bulk water. However, the concentration was lower than that of denitrification-based N_2O . More importantly, the nitrifier denitrification-based N_2O production in the plastsphere was higher than that in bulk water. This further strengthens our claim that the plastsphere is an overlooked source of N_2O production. We have now added this figure in the revised manuscript as Supplementary Fig. 9, and also deleted the sentence regarding "we simulated in situ conditions" in the Main Text.

As the aim of this study is to explore the denitrification-based N_2O production (including bacterial, fungal and chemical sources) in the plastsphere and bulk water, much attention on nitrification-based N_2O production will distract readers from the scope. Nevertheless, we have now added several statements in the revised manuscript to describe the supplemented results, and also to discuss the limitations of this study regarding nitrifier N_2O source. We believe that although our study focuses on N_2O production from denitrification pathways, nitrification process as implied from this study should also be considered in future studies on plastsphere. This will complement the current study in providing additional information for better understanding the role of aquatic plastsphere in biogeochemical cycling.

"It is worth noting that nitrification process in the plastsphere and bulk water, including hydroxylamine oxidation and nitrifier denitrification^{41,65}, may also contribute to N_2O production. We further conducted a lab-scale experiment under in situ dissolved oxygen concentration ($\sim 1.2 \text{ mg L}^{-1}$, hypoxic conditions) to explore nitrifier potential in N_2O production in the plastsphere and bulk water (Supplementary Fig. 9). There was no significant difference observed between groups with and without adding $80\mu\text{M}$ of allylthiourea (a nitrification inhibitor⁶⁶), indicating that N_2O production via hydroxylamine oxidation was minimal under in

situ conditions. By contrast, nitrifier denitrification was likely a potential contributor in the plastisphere and bulk water (Supplementary Fig. 9b), but the concentration was lower than that of denitrification-based N_2O . More importantly, the nitrifier denitrification-based N_2O production in the plastisphere was higher than in bulk water (Supplementary Fig. 9). This further supports that the plastisphere is an overlooked source of N_2O production. Although in-depth discussions on the nitrifier-related sources of N_2O are beyond the scope of this study, we believe that it is urgent to investigate nitrification and associated N_2O dynamics in future research, which will deepen the understanding of the role of aquatic plastisphere in biogeochemical cycling." (Page 18 Lines 356-372).

".....aiming to ~~simulate in situ conditions~~ and explore the basal capacities of denitrification....." (Page 22 Line 439).

4. The author's claim that sufficient organic carbon was present is debatable – as the experiments are conducted under batch conditions (and the experiment follows the depletion of the originally present nitrate – which would already consume some of the original organic carbon that is available) – while in situ substantive fluxes of carbon would be replenish consumed carbon. Was there sufficient carbon for denitrification? The authors state the initial (in situ) concentrations; I suggest that authors also add a measurement of the carbon at the addition of the nitrate spike and the remaining organic carbon at the end of the batch experiments. This condition, again, matters as carbon availability can have a substantive effect on N_2O production.

Answer:

Thanks for this comment. We would like to mention that in this study, the lab-scale experiments were not conducted in batch conditions (one experiment after

another), but were conducted in parallel. Each experiment, including the determinations of denitrification rate, N₂O reduction ratio, and N₂O concentration and isotopocules, was respectively conducted with (sterile) fresh estuarine water taken from the estuaries. Thus, organic matter had not been likely consumed before experiments. To avoid confusion, we have revised the manuscript and the figure to make this point more clearly.

"Each lab-scale experiment was conducted in parallel, and variations in the concentrations of organic matter during the lab-scale experiment are shown in Supplementary Fig. 10." (Page 22 Lines 440-442, and also in Supplementary Fig. 1 and 10).

Response to comments Fig. 2 (now updated in the Supporting Information, Supplementary

Fig. 1). Schematic diagram of experimental design.

To further confirm if it is sufficient for denitrification, we have added another measurement to investigate the changes in dissolved organic matters (DOM) under anoxic and hypoxic conditions (please see figure below). The initial concentration of DOM was 18.26 mg L^{-1} in the estuarine water. During the 30h incubation, the concentrations of DOM decreased along with the incubation time, regardless of the anoxic and hypoxic conditions. At the 24h, DOM remained around $9\sim 13 \text{ mg L}^{-1}$ in the plastisphere and bulk water groups; while the concentrations decreased to $2\sim 7 \text{ mg L}^{-1}$ at the 30h. This indicates that the in situ organic matters are sufficient for denitrification during the previous 24h experiment.

Response to comments Fig. 3 (now included in the Supporting Information, **Supplementary**

Fig. 10). A supplemented 30h experiment showing the changes in the concentrations of dissolved organic matters (DOM) in the plastisphere and bulk water groups under both anoxic (a) and hypoxic (b) conditions. Anoxic conditions were achieved by purging with He gas (>99.99%); Hypoxic conditions were achieved by purging with He (95%) and O_2 (5%) gases. PE plastisphere was selected as a representation of plastisphere group. Concentrations of DOM were measured with a TOC analyzer (TOC-L, SHIMADZU, Japan) after filtering $0.22\mu\text{m}$ membrane.

Reviewer #3 Comments:

5. All of my previous comments and concerns have been sufficiently addressed by the authors.

Answer:

We thank the reviewer again for taking the time to assess our study. The manuscript has been substantially improved after incorporating the reviewer's comments.

References

1. Ouyang, L.; Thamdrup, B.; Trimmer, M. Coupled nitrification and N₂ gas production as a cryptic process in oxic riverbeds. *Nat. Commun.* 2021; 12 (1).
2. Lehnert, N.; Dong, H. T.; Harland, J. B.; Hunt, A. P.; White, C. J. Reversing nitrogen fixation. *Nat. Rev. Chem.* 2018; 2 (10):278-289.
3. Kuypers, M. M. M.; Marchant, H. K.; Kartal, B. The microbial nitrogen-cycling network. *Nat. Rev. Microbiol.* 2018; 16 (5):263-276.

REVIEWERS' COMMENTS

Reviewer #2 (Remarks to the Author):

The authors have adequately addressed my final comments to the manuscript.

Replies and Explanations to Comments

Reviewer #2 Comments:

1. The authors have adequately addressed my final comments to the manuscript.

Answer:

We thank the reviewer for the positive feedback, and the study has been substantially improved after incorporating the reviewer's comments.